# Genome-wide CRISPR screens identify PKMYT1 as a therapeutic target in pancreatic ductal adenocarcinoma

Simin Wang [ID][1], Yangjie Xiong[1], Yuxiang Luo[1], Yanying Shen[2], Fengrui Zhang[3], Haoqi Lan[1], Yuzhi Pang[1], Xiaofang Wang [ID][1], Xiaoqi Li[4], Xufen Zheng[1], Xiaojing Lu[1], Xiaoxiao Liu[1], Yumei Cheng [ID][1], Tanwen Wu [ID][1], Yue Dong[1], Yuan Lu[3], Jiujie Cui[5], Xiaona Jia [ID][1], Sheng Yang [ID][6], Liwei Wang [ID][5][✉] & Yuexiang Wang [ID][1][✉]

## Abstract

**Pancreatic ductal adenocarcinoma (PDAC) is a devastating disease with an overall 5-year survival rate of <12% due to the lack of effective treatments. Novel treatment strategies are urgently needed. Here, *PKMYT1* is identified through genome-wide CRISPR screens as a non-mutant, genetic vulnerability of PDAC. Higher PKMYT1 expression levels indicate poor prognosis in PDAC patients. PKMYT1 ablation inhibits tumor growth and proliferation in vitro and in vivo by regulating cell cycle progression and inducing apoptosis. Moreover, pharmacological inhibition of PKMYT1 shows efficacy in multiple PDAC cell models and effectively induces tumor regression without overt toxicity in PDAC cell line-derived xenograft and in more clinically relevant patient-derived xenograft models. Mechanistically, in addition to its canonical function of phosphorylating CDK1, PKMYT1 functions as an oncogene to promote PDAC tumorigenesis by regulating PLK1 expression and phosphorylation. Finally, *TP53* function and PRKDC activation are shown to modulate the sensitivity to PKMYT1 inhibition. These results define PKMYT1 dependency in PDAC and identify potential therapeutic strategies for clinical translation.**

**Keywords** Pancreatic Ductal Adenocarcinoma; CRISPR Screens; PKMYT1
**Subject Categories** Cancer; Digestive System

## Introduction

Pancreatic ductal adenocarcinoma (PDAC), accounting for more than 90% of pancreatic cancers, is a highly aggressive malignancy with an overall 5-year survival rate of <12% due to the lack of effective treatments and late-stage diagnosis (Siegel et al, 2023). PDAC is predicted to become the third leading cause of cancer mortality by 2030 (Siegel et al, 2023). Currently, the treatment options for PDAC are limited. This disease is usually diagnosed at a late stage, which eliminates the opportunity for curative surgical resection (Adamska et al, 2017). Chemotherapy is the most frequently used approach for PDAC treatment and has limited effects (Conroy et al, 2011). Most treatments are palliative, focused on symptom relief and life extension (Hammel et al, 2016). Therefore, there is an urgent need to understand the molecular mechanisms of PDAC and develop more effective therapeutic strategies.

The genomic landscape of PDAC has provided a rich source of therapeutic target identification (Network, 2017; Waddell et al, 2015). The current actively investigated targets are KRAS and mutant-BRCA (Golan et al, 2019; Strickler et al, 2023). Novel treatment strategies are urgently needed. CRISPR-Cas9 screen is a powerful approach for identifying genes that are essential for tumor growth and proliferation, serving as a powerful source of non-mutant target discovery (Bock et al, 2022; Shalem et al, 2014).

The protein kinase membrane-associated tyrosine/threonine 1 (PKMYT1), encoded by the *PKMYT1* gene, is one of the three WEE kinase family members (Asquith et al, 2020; Lee and Yang, 2001). The main downstream target of PKMYT1 is the cyclin-dependent kinase 1 (CDK1)-cyclin B1 complex, also called mitotic-promoting factor (MPF). It phosphorylates Thr14 and Tyr15 of CDK1 (Lee and Yang, 2001), keeping the complex trapped in the cytoplasm and inhibited (Lescarbeau et al, 2016) until the appropriate time for progression from G2 phase to the mitosis (Long et al, 2020). Recently, bioinformatics analysis has shown that overexpression of PKMYT1 potentially correlates with poor prognosis in clear cell renal cell carcinoma (Chen et al, 2020). PKMYT1 is a candidate predictive biomarker of acquired resistance to the WEE1 kinase inhibitor in breast cancer (Lewis et al, 2019). However, the role and mechanism of PKMYT1 in PDAC tumorigenesis remain unknown.

[1]CAS Key Laboratory of Tissue Microenvironment and Tumor, Shanghai Institute of Nutrition and Health, Chinese Academy of Sciences, University of Chinese Academy of Sciences, 200031 Shanghai, China. [2]Department of Pathology, Ren Ji Hospital, School of Medicine, Shanghai Jiao Tong University, 200127 Shanghai, China. [3]Department of Gynecology, Obstetrics and Gynecology Hospital, Fudan University, 200011 Shanghai, China. [4]Department of Gastrointestinal Surgery, Ren Ji Hospital, School of Medicine, Shanghai Jiao Tong University, 200127 Shanghai, China. [5]Department of Oncology, Ren Ji Hospital, School of Medicine, Shanghai Jiao Tong University, 200127 Shanghai, China. [6]Department of Oncology, Fujian Medical University Union Hospital, 350001 Fuzhou, Fujian, China. ✉E-mail: liweiwang@shsmu.edu.cn; yxwang76@sibs.ac.cn

To identify potential therapeutic candidates for pancreatic cancer, we performed whole-genome CRISPR/Cas9 loss-of-function screens. PKMYT1 was identified as a pivotal tumor vulnerability in PDAC. Higher expression levels indicate poor prognosis in PDAC patients. PKMYT1 ablation inhibits tumor growth and proliferation in vitro and in vivo by regulating cell cycle progression and inducing apoptosis. In addition to its canonical function of phosphorylating CDK1, it interacts with PLK1 in PDAC. PKMYT1 functions as an oncogene to promote PDAC tumorigenesis by regulating PLK1 expression and phosphorylation. PRKDC activation and p53 pathway modulate the sensitivity of PDAC to PKMYT1 inhibition. Moreover, RP-6306, an orally bioavailable inhibitor, is found to effectively inhibit PKMYT1 selectively in PDAC. Our findings identify a novel vulnerability in PDAC cells: PKMYT1, which functions as an oncogene, can be inhibited to provide additional therapeutic strategies for treating PDAC.

## Results

### Genome-wide CRISPR screens identify PKMYT1 as a genetic vulnerability of PDAC cells

To systemically identify the genetic vulnerabilities in PDAC, we first conducted whole-genome CRISPR screens using the human knockout one-vector lentiviral system GeCKOv2 (genome-scale CRISPR knockout v2) (Sanjana et al, 2014), which contained 123,411 unique sgRNAs targeting 19,050 genes (6 sgRNAs per gene) and 1000 nontargeting negative control sgRNAs, in two representative human pancreatic adenocarcinoma cell lines, Pa-Tu-8988T and YAPC, established from advanced pancreatic adenocarcinoma. After 14 population doublings (PD14), genomic DNA encompassing the sgRNA, with ~300× coverage, was extracted and assessed by PCR amplification and next-generation sequencing (Fig. EV1A; Dataset EV1). As expected, the sgRNA distributions were significantly shifted between PD14 with PD0, indicating that the screens functioned as designed and that the sgRNA diversity was significantly reduced in the surviving cells (Fig. EV1B). For each sgRNA, we calculated the CRISPR score (CS) to represent its dropout degree. Considering the off-target effects of sgRNA, we defined the median score (med.CS) of all sgRNAs targeting one gene at the PD14 time point as a representative measure to characterize the effects of the gene on cell proliferation; therefore, there were at least three effective sgRNAs (Dataset EV1). In addition, we also calculated the β score using the MAGeCK-MLE algorithm (Li et al, 2014) to improve the accuracy of the screens. The med.CS was strongly correlated with the β score (Fig. EV1C). A split-library approach utilizing paired human whole-genome sgRNA libraries (denoted library A and library B) was used, with ~3 sgRNAs per gene in each library. A correlation in the dependency scores (med.CS) was observed between the two libraries (Fig. 1A) as well as between the two PDAC cell lines (Fig. 1B). The screen datasets were then merged for subsequent analysis to improve the statistical power.

Theoretically, genes whose deletion causes cell proliferation defects should preferentially be observed with lower value, as preliminary candidates. Then, we compared three sets of common essential genes and two sets of nonessential genes, finding that the

performance of the screens was in line with expectations (Fig. EV1D) (Hart et al, 2015; Tsherniak et al, 2017). To identify dropout hits for viability in cells, we rank-ordered the genes by the med.CS from the most negative to the most positive. Functional annotation with the DAVID tool revealed that most of the depleted sgRNAs targeted genes in essential pathways such as Ribosome and Cell cycle (Fig. 1C). The overlap in highly depleted genes and functional gene categories between the two cell lines indicates that screens can identify essential genes and that enrichment analysis of the depleted sgRNAs pinpoints key functional gene categories in negative selection screens. In addition, these results also suggested several gene addiction is cell-specific (Fig. 1D). For the sake of seeking rational targets for exploitation in PDAC, we subsequently compared our screens with two other screens in gastrointestinal stromal tumors (GIST) (Fig. 1E; Dataset EV1). The hits were prioritized by eliminating previously identified essential genes that had been shown to be critical in most cell lines (Hart et al, 2015; Tsherniak et al, 2017) and incorporating components of the druggable genome (Griffith et al, 2013; Wagner et al, 2016), which included known targets of existing drug compounds ("drugged" genes) and genes that belong to categories predicted to be druggable ("druggable" genes) (Li et al, 2019). Finally, we revealed several genes including *PKMYT1*, *SDHC* and *COX7B*, that significantly impeded growth. However, we were most interested in the targets associated with patient prognosis. Given this criterion, *PKMYT1* was identified as a potentially selective target (Figs. 1E and 2; Appendix Fig. S1).

### High PKMYT1 expression correlates with poor patient survival in PDAC

To further delineate the potential role of PKMYT1 in PDAC, we assessed PKMYT1 expression in human PDAC specimens. Immunohistochemical (IHC) staining with validated antibody against PKMYT1 demonstrated high PKMYT1 expression (36%) on a human tissue microarray consisting of 75 patients with PDAC (Fig. 2A). The high *PKMYT1* expression pattern in PDAC was externally validated in the TCGA-PAAD cohort (Fig. 2B). The expression level of *PKMYT1* progressively increased with gradually decreased degree of tumor differentiation (Fig. 2C). Analyses with three independent cohorts also confirmed that higher *PKMYT1* expression correlates with a worse prognosis in PDAC patients (Fig. 2D) (Data ref: Moffitt et al, 2015; Data ref: Stratford et al, 2017). Together, these findings prompt us to study further to elaborate the role of PKMYT1 in PDAC tumorigenesis.

### PKMYT1 knockout inhibits PDAC tumor growth and proliferation

Next, the biological function of PKMYT1 was investigated using various human PDAC models. CRISPR-mediated *PKMYT1* knockout (Fig. 3A) in PDAC cell lines (Pa-Tu-8988T and YAPC) and primary cultured cell (PDAC-CN1) dramatically inhibited cell growth and proliferation in both short-term (Fig. 3B) and long-term (Fig. 3C) assays (Dataset EV2; Table EV1). PKMYT1 knockout also severely impaired the three-dimensional anchorage-independent growth (Fig. 3D). To further test the role of PKMYT1 in PDAC, we extended these findings to another primary cultured cell (PDAC-CN2) and three additional PDAC cell lines, including two *KRAS* wild-type lines (BxPc3

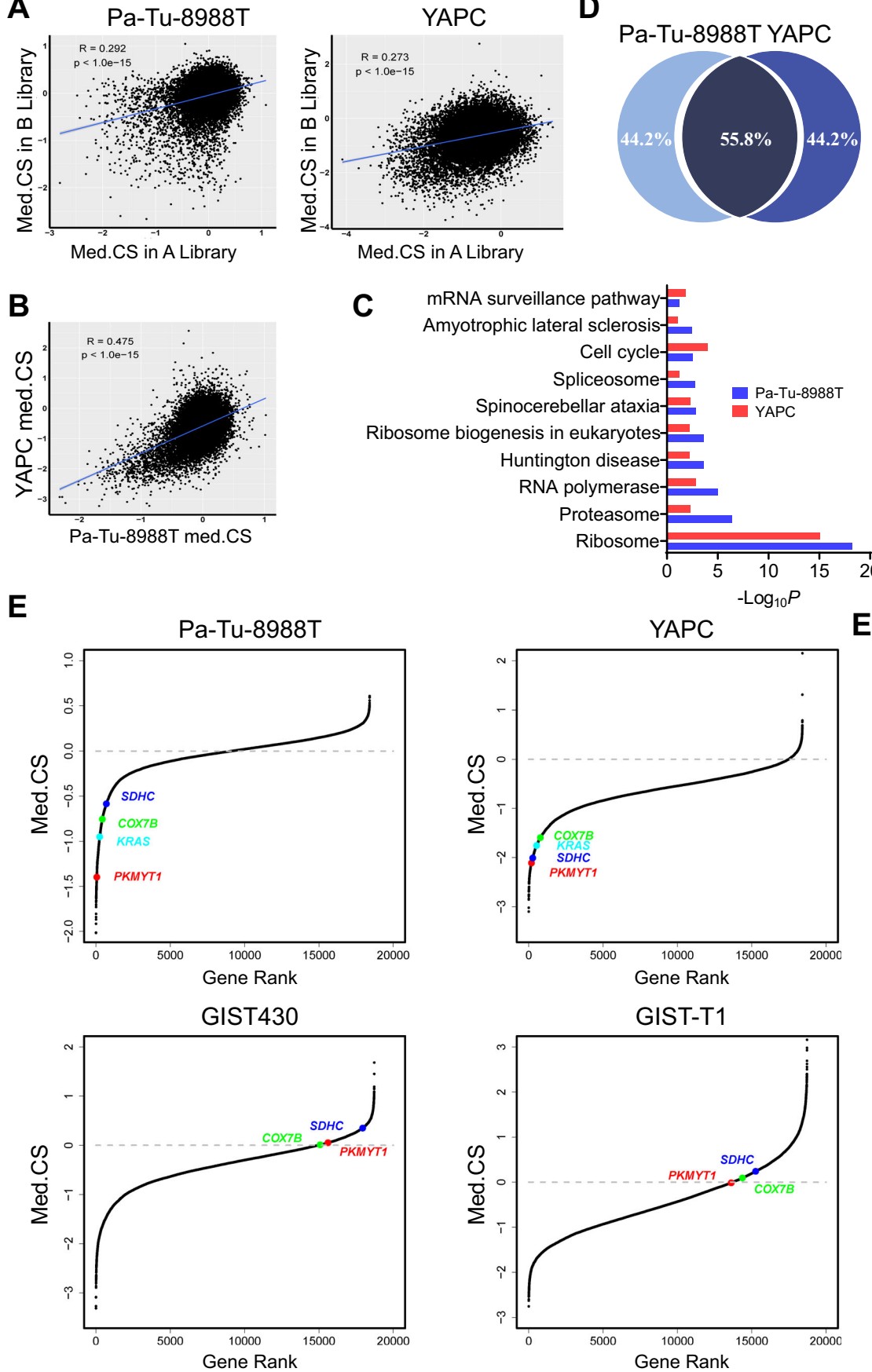

**Figure 1. Identification of PDAC dependencies through genome-wide CRISPR loss-of-function screens.**

(A) Correlation of med.CS between sgRNA libraries A and B. The data were derived from screens in the Pa-Tu-8988T (left) and YAPC (right) cell lines. Unpaired *t* test; pearson correlation coefficient was used. (B) Significant correlation of med.CS between the Pa-Tu-8988T and YAPC cell lines. Unpaired *t* test; pearson correlation coefficient was used. (C) DAVID analysis shows that the top 200 genes (~1%) identified as essential in PDAC screens are significantly enriched in fundamental cellular processes. Fisher's exact test. (D) Venn diagram shows the overlap among the top 1000 dropout genes (~5%) identified in the two screens. (E) Rank-ordered med.CS for each gene from screens in the Pa-Tu-8988T, YAPC, GIST430, and GIST-T1 cell lines. The *PKMYT1* gene is indicated by the red point. The *KRAS, COX7B* and *SDHC* are indicated with color cyan, color green, and color blue, respectively.

and SNU-324). Consistent with the above data, PKMYT1 ablation reduced cell growth and proliferation (Fig. EV2A,B). To determine whether the inhibition of cell proliferation is manifested in vivo, we generated three types of xenografts (Pa-Tu-8988T, YAPC, and PDAC-CN1 with or without *PKMYT1* KO) in nude mice. In vivo experiments showed that *PKMYT1* KO markedly attenuated tumor growth although the tumors harbored oncogenic *KRAS* mutations (Fig. 3E–G). Collectively, these results demonstrate that PKMYT1 knockout reduces the PDAC cell growth and proliferation in vitro and in vivo.

Functionally, PKMYT1, a serine/threonine kinase, inhibits CDK1 activity and prevents its association with cyclin B through phosphorylation of CDK1 Tyr15 and Thr14 residues. In contrast to the Tyr15 residue of CDK1, PKMYT1 is the only one kinase found to phosphorylate CDK1 on the Thr14 residue (Mueller et al, 1995). As expected, PKMYT1 knockout downregulated Thr14 phosphorylation of CDK1 in PDAC (Fig. EV2C). Reintroduction of a sgRNA-resistant *PKMYT1* transgene protects cells from the depletion of endogenous PKMYT1 by sgRNA, whereas expression of sgRNA-resistant PKMYT1 with the N238A mutation, which encodes a protein with kinase dysfunction (Gallo et al, 2022), does not (Fig. EV2D,E). We concluded that the protein kinase activity of PKMYT1 participates, at least partially, in the regulation of tumor proliferation. Considering that CDK1 is a master regulator of the cell cycle, whose activation is essential for entry into mitosis, we analyzed all phases of the cell cycle (cell cycle distribution) by flow cytometry. PKMYT1 ablation inhibited cell cycle progression through the G2/M checkpoint, increasing the proportion of cells in G2/M phase (Figs. 3H and EV2F). Accumulation of γH2AX was induced by *PKMYT1* knockout (Figs. 3I and EV2C), as pan-nuclear γH2AX-positive cells were observed via immunofluorescence staining, consistent with a previous description of both WEE kinase family member inhibition (Aarts et al, 2012; Gallo et al, 2022) and $Cdk1^{AF/AF}$ mouse (Szmyd et al, 2019), which is in line with expectations that loss of kinase activity interferes with the G2/M checkpoint, driving cells into mitosis prematurely (Asquith et al, 2020; Chow and Poon, 2013) and causing genome instability (Gallo et al, 2022). In addition, flow cytometric analysis also showed that knockout of *PKMYT1* increased apoptosis, while *PKMYT1* expression correlates with the proliferative index indicated by *MKI67* (Figs. 3J and EV2G,H). Transcriptome profiling revealed that the mitotic spindle, G2/M checkpoint and apoptosis signatures were enriched in PKMYT1-ablated PDAC cells (Fig. EV2I; Dataset EV3). Interestingly, PKMYT1 dropout sensitized pancreatic cancer cells to gemcitabine (Fig. EV2J).

## Antitumor activity of pharmacological inhibition of PKMYT1

To assess the therapeutic potential of pharmacological inhibition of the oncogenic PKMYT1, we tested the sensitivity of PDAC models

to small-molecule inhibitors of PKMYT1. RP-6306 is the first potent, selective, and orally bioavailable PKMYT1 inhibitor and the crystal structure of RP-6306 binding directly to PKMYT1 has been demonstrated (Gallo et al, 2022; Szychowski et al, 2022) (Fig. 4A; Dataset EV4). RP-6306 treatment caused reduction of CDK1 phosphorylation (Fig. 4B) as well as hyperphosphorylation of PKMYT1 indicated by drastically decreased mobility (Chow and Poon, 2013) (Fig. 4B), even when the drug concentration was lower than its $IC_{50}$ value. It also induced cell cycle arrest and promoted DNA lesion accumulation, accompanied by increased apoptosis (Fig. 4C–E) in a concentration-dependent manner.

The in vivo antitumor activity of RP-6306 was evaluated in PDAC cell line-derived xenograft (CDX) models and in more clinically relevant PDAC patient-derived xenograft (PDX) models (Table EV1). In the PDAC cell line xenograft models, RP-6306 stabilized tumor growth when given as a single agent, showing marked effects in vivo (Fig. 5A–C). We next further tested the antitumor activity of RP-6306 in PDX (PDAC-PDX2) models with PKMYT1 expression and *KRAS* G12D mutation. In this model, RP-6306 reduced the tumor volume by 50% after 15 days of therapy (Fig. 5D–F). No evidence of liver toxicity was detected by monitoring serum levels of alanine aminotransferase (ALT) and aspartate aminotransferase (AST) in the mice treated with RP-6306 (Fig. 5G,H). Routine peripheral blood tests showed no significant differences in the RP-6306-treated mice (Fig. EV3A,B). The data show that RP-6306 treatment results in tumor regression without overt toxicity or weight loss.

In addition to RP-6306, other broad-spectrum kinase inhibitors, such as the tyrosine kinase inhibitor dasatinib and the pyridopyrimidine derivative PD0166285 (Platzer et al, 2018; Rohe et al, 2013) also regulate PKMYT1 activity and reduce CDK1 phosphorylation at Thr14 (Appendix Figs. S2A and S3A). Decreased cell viability in a range of PDAC cell models with PKMYT1 expression and phenocopies of PKMYT1 excision in regard to DNA damage, apoptosis and cell cycle progression were observed when treated with these inhibitors (Appendix Figs. S2B–E and S3B–D; Dataset EV4). Further optimization of the structure and specificity of these broad-spectrum inhibitors is expected to obtain more promising clinical drugs.

## PKMYT1 interacts with PLK1 and PLK1 mediates PKMYT1 oncogenic roles in PDAC

To investigate whether there are other underlying noncanonical mechanisms of PKMYT1 in the pancreatic cancer context, Flag-tagged PKMYT1 was used as bait to identify proteins that potentially interact with PKMYT1 through tandem mass spectrometry with co-immunoprecipitation. CDK1, a well-known substrate, was identified in the proteomic screen (Fig. 6A,B). The top-hit proteins with the highest confidence that potentially interact

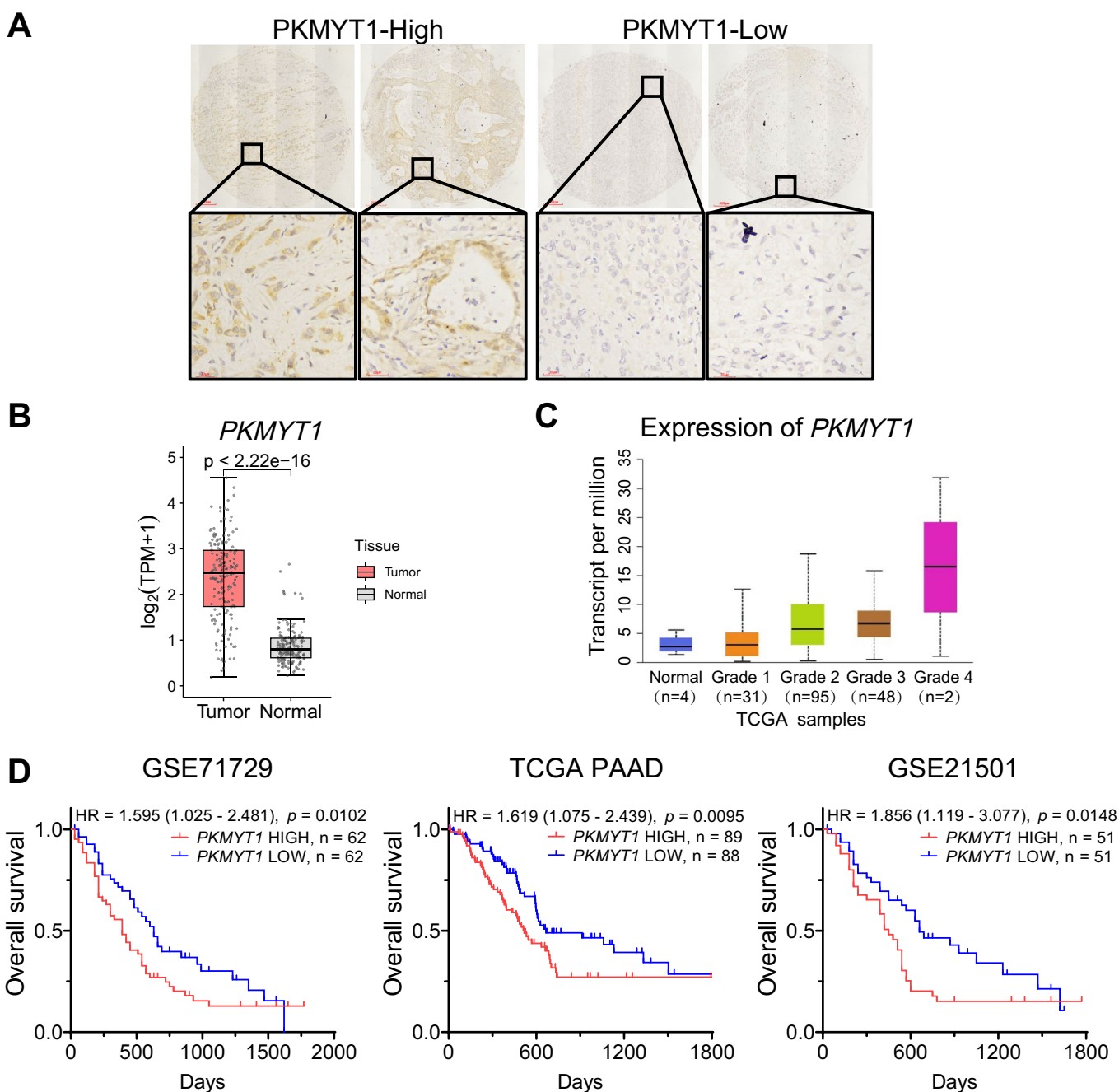

**Figure 2. High expression of PKMYT1 in PDAC and its clinical significance.**

(A) Immunohistochemical analysis in a human tissue microarray consisting of patients ($n = 75$) with PDAC. Representative pictures of expression levels are shown. High indicates expression higher than or equal to tumor-adjacent normal tissues. Low indicates expression lower than tumor-adjacent normal tissues. Scale bars: 300 μm and 30 μm, respectively. (B) PKMYT1 expression in human pancreatic cancer tissues ($n = 182$) compared with noncancerous tissues ($n = 169$) from the TCGA-PAAD (pancreatic adenocarcinoma) cohort. The low bound, centerline, and upper bound of boxplot represent the first quartile, the median, and the third quartile of data, respectively; the upper and lower whiskers extend to the largest and smallest value within 1.5 times of the interquartile range. Unpaired $t$ test; pearson correlation coefficient was used. (C) The expression level of PKMYT1 progressively increases as the degree of tumor differentiation decreases. Grade 1, well-differentiated (low grade, $n = 31$); Grade 2, moderately differentiated (intermediate grade, $n = 95$); Grade 3, poorly differentiated (high grade, $n = 48$); Grade 4, undifferentiated (high grade, $n = 2$). The low bound, centerline, and upper bound of boxplot represent the first quartile, the median, and the third quartile of data, respectively; the upper and lower whiskers extend to the largest and smallest value. (D) Kaplan–Meier curves for overall survival show that PDAC patients with high PKMYT1 expression have worse prognoses than those with low PKMYT1 expression in GEO datasets GSE71729 ($n = 124$), GSE21501 ($n = 102$) and the TCGA-PAAD dataset ($n = 177$). HR hazard ratio, with 95% confidence interval. Chi-square test. Source data are available online for this figure.

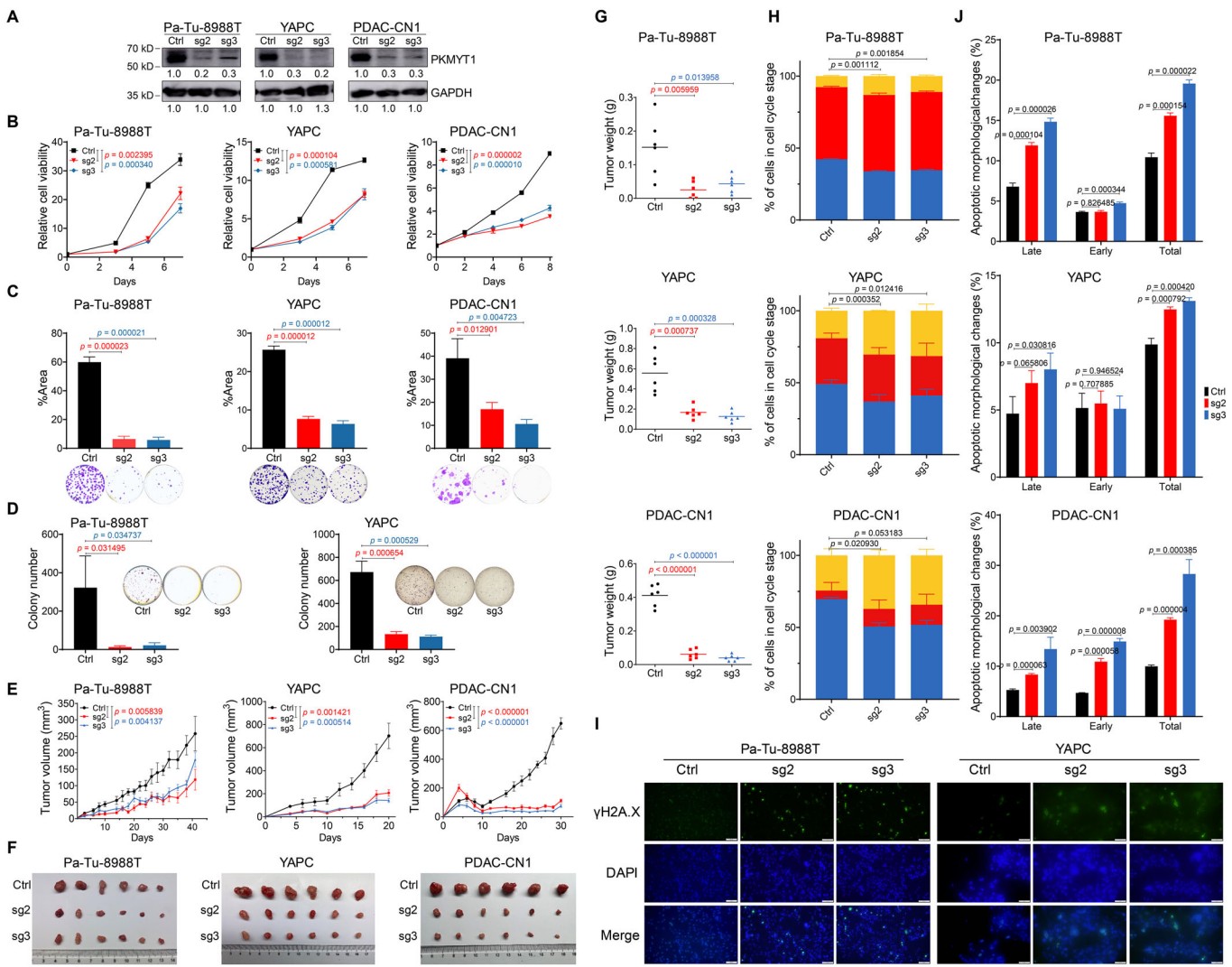

**Figure 3. *PKMYT1* knockout inhibits tumor growth and proliferation in PDAC cell models in vitro and PDAC xenograft models in vivo.**

(**A**) CRISPR-mediated *PKMYT1* knockout (sg2 and sg3) decreases PKMYT1 protein as indicated by western blotting. (**B**) CRISPR-mediated *PKMYT1* knockout reduces the viability of Pa-Tu-8988T, YAPC and PDAC primary cultured cell PDAC-CN1, as assessed by the CellTiter-Glo viability assay. The error bars indicate the mean ± s.d. of three replicates; unpaired *t* test. (**C**) Quantification and representative plates of crystal violet staining assays show that *PKMYT1* depletion suppresses the proliferation of PDAC cells. The error bars indicate the mean ± s.d. of three replicates; unpaired *t* test. (**D**) Quantification and representative plates of soft-agar assays show that *PKMYT1* depletion inhibits the anchorage-independent growth of PDAC cells. The error bars indicate the mean ± s.d. of three replicates; unpaired *t* test. (**E–G**) *PKMYT1* deletion attenuates the growth of Pa-Tu-8988T, YAPC and PDAC-CN1 xenografts in nude mice. Growth curves (**E**), representative photo images (**F**) and tumor weight (**G**) of transplanted tumors are shown. The error bars indicate the mean ± s.d. of six replicates (*n* = 6 per group); unpaired *t* test. (**H**). Analysis of the effect of *PKMYT1* on the PDAC cell cycle by flow cytometry. *PKMYT1* knockout causes G2/M arrest in cells. The error bars indicate the mean ± s.d. of three replicates; unpaired *t* test. (**I**) Immunofluorescence assay shows that *PKMYT1* knockout increases DNA damage in the nucleus, as indicated by pan-nuclear H2A.X phosphorylation. Scale bars: 100 μm in Pa-Tu-8988T, 50 μm in YAPC. (**J**) Analysis of the effect of *PKMYT1* on PDAC cell apoptosis by flow cytometry. Knockout of *PKMYT1* promotes apoptosis (*n* = 3 per group). The error bars indicate the mean ± s.d. of three replicates; unpaired *t* test. Ctrl control. (**A, B, H–J**) represent data from three biological replicates, (**C, D**) represent data from two biological replicates. Source data are available online for this figure.

with PKMYT1 were further studied. The interaction between PLK1 and PKMYT1 was validated in Pa-Tu-8988T (endogenous PLK1/PKMYT1 interacts with exogenous PKMYT1/PLK1) and in three additional PDAC cell lines (endogenous PLK1 interacts with endogenous PKMYT1, and vice versa) (Fig. EV4A–E). *PKMYT1* knockout or pharmacological inhibition downregulated the expression of PLK1 in PDAC cell lines and primary cultured cell (Figs. 4B and 6C). Consistent with this finding, overexpression of PKMYT1 increased the PLK1 protein expression (Fig. 6D). Cells were treated

with cycloheximide (CHX) to inhibit protein biosynthesis (Sharma et al, 2021). Knockout of PKMYT1 in PDAC cells caused a significant decrease in the half-life of PLK1 (Fig. 6E). These results indicated that PKMYT1 mediates PLK1 degradation in PDAC cells. Furthermore, the effect of PKMYT1 on PLK1 could be blocked by the proteasome inhibitor MG132 (Fig. 6F). These results show that PKMYT1 regulates the proteasome-mediated PLK1 degradation.

Analysis by online website UALCAN revealed a positive correlation between *PKMYT1* and *PLK1* expression in pancreatic

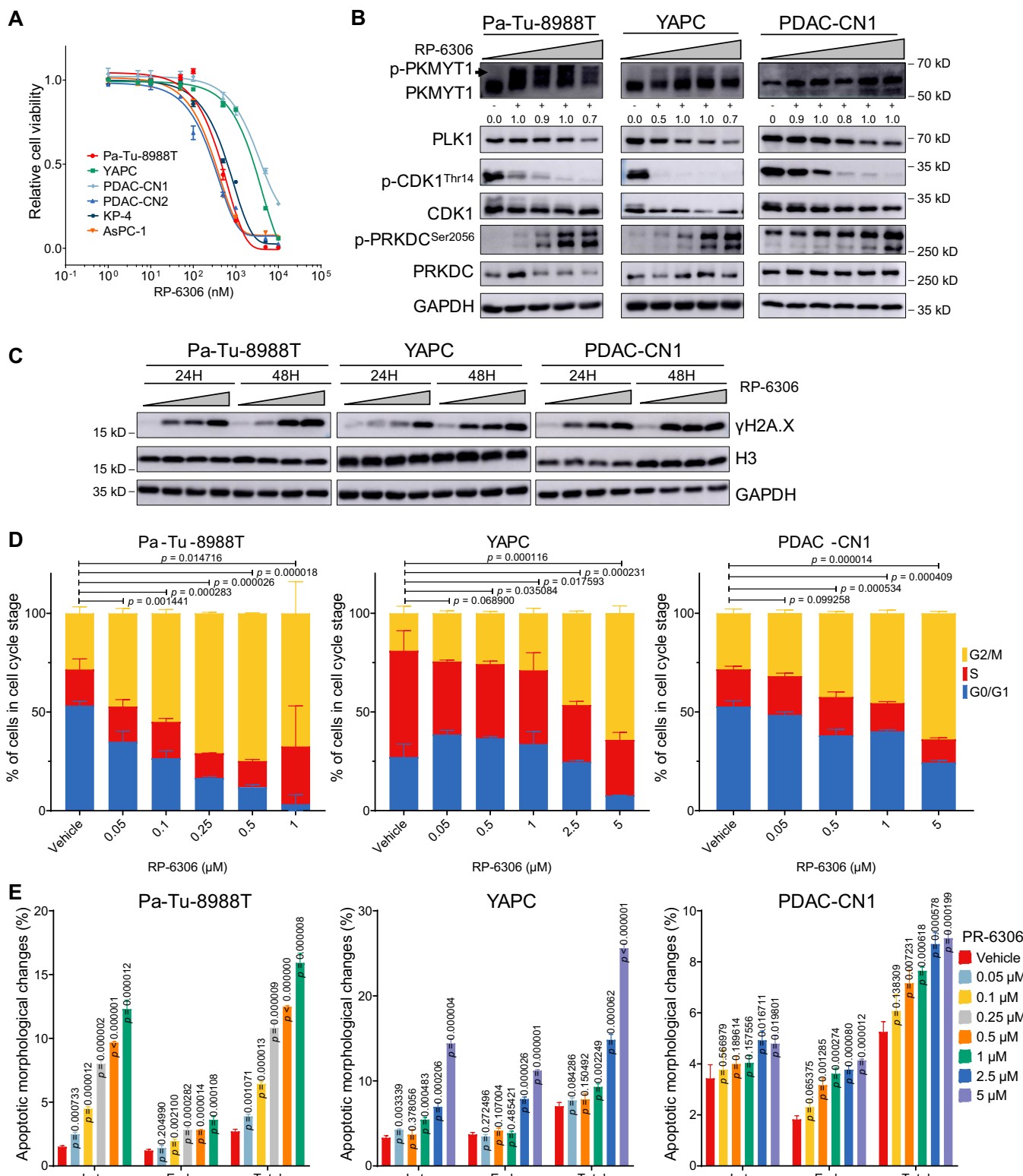

cancer cohorts (Fig. EV4F). These data demonstrate that PKMYT1 has a strong regulatory relationship with PLK1, suggesting that they may be functional partners in PDAC. High *PLK1* expression has astrong positive relationship with the occurrence of tumors, and PLK1 overexpression has been implicated in tumorigenesis

(Degenhardt and Lampkin, 2010). Higher *PLK1* expression was associated with a worse grade of PDAC (Fig. EV4G). We further demonstrate that *PLK1* knockout leads to cell proliferation defects (Fig. 6G), G2/M arrest, an increased proportion of cells in the G2/M phase (Fig. 6H), and apoptosis (Fig. 6I), phenocopying the

**Figure 4. RP-6306, a PKMYT1 inhibitor, suppresses tumor growth and proliferation in multiple PDAC cell models in vitro.**

(A) RP-6306 is an effective inhibitor in PKYMY1-expressing PDAC cell lines. The error bars indicate the mean ± s.e.m. of three replicates. (B) RP-6306 treatment increases the hyperphosphorylation of PKMYT1 (indicated by drastically decreased mobility), inhibits its kinase activity (as reduction of CDK1 phosphorylation), decreases PLK1 protein expression and activates PRKDC by increasing its Ser2056 phosphorylation. The mobility of PKMYT1 was made qualitative and relative quantitative assessments. (C) RP-6306 treatment increases γH2A.X accumulation in a concentration- and time-dependent manner. (D) RP-6306 treatment causes cell cycle arrest, as determined by flow cytometry. The error bars indicate the mean ± s.d. of three replicates; unpaired t test. (E) RP-6306 treatment increases apoptosis, as determined by flow cytometry. The error bars indicate the mean ± s.d. of three replicates; unpaired t test. Concentrations of RP-6306 used (B, C) are listed in the "Methods". (B, D, E) Represent data from two biological replicates. Source data are available online for this figure.

effects of PKMYT1 ablation except the accumulation of DNA damage (Fig. EV4H,I). In addition, enforced PLK1 expression significantly attenuated growth and proliferation inhibition properties in PKMYT1-ablated PDAC cells (Fig. 6J).

Then, to investigate whether PKMYT1 phosphorylates PLK1, an in vitro incubation of PLK1 immunoprecipitates with purified recombinant PKMYT1 enzyme in the presence of ATP was performed to confirm the direct phosphorylation of PLK1 by PKMYT1 (Fig. EV4J) (Gelot et al, 2023; Moon et al, 2017). As *PKMYT1* gene encodes a member of the serine/threonine protein kinase family, we observed PLK1 phosphorylation at pan-Ser/Thr sites when incubated with PKMYT1. The phosphorylation was markedly reduced when PKMYT1 inhibitor was added to the reaction system (Fig. EV4J). Similarly, as PKMYT1 regulates the phosphorylation of CDK1 Thr14 and Tyr15, we focused on two reported residues Thr210 and Tyr217, which are functionally important (Caron et al, 2016; Macurek et al, 2008). We identified a PKMYT1 phosphorylation site on PLK1 Tyr217. The phosphorylation of PLK1 Tyr217 was markedly increased when incubated with PKMYT1 compared with PKMYT1 kinase inhibited by RP-6306 (Fig. EV4J). Next, to determine if the phosphorylation site is essential for the oncogenic function of PKMYT1, different mutants were constructed. Consistently, phospho-mimicking PLK1 Tyr217-Asp mutant but not Tyr217Phe mutant (which can't be phosphorylated) overexpression attenuated the growth and proliferation inhibition properties in PKMYT1-ablated PDAC cells (Fig. EV4K,L). Together, our results show that PKMYT1 phosphorylates PLK1 at Tyr217 and other potential Ser/Thr sites. Tyr217 phosphorylation is involved in the oncogenic function of PKMYT1, although we can not rule out whether other Ser/Thr residues are also involved. The detailed mechanisms between PKMYT1 and PLK1 merit further investigation.

Collectively, our functional studies, together with biochemical data, show that PKMYT1 promotes PDAC tumorigenesis through regulating PLK1 expression and phosphorylation (Appendix Fig. S4).

## PRKDC activation promotes PKMYT1 inhibition-induced γH2AX accumulation and cytotoxicity in PDAC

That PLK1 knockout couldn't mimic the γH2AX accumulation phenotype caused by PKMYT1 inhibition led us to investigate other factors involved in DNA damage response (DDR) (Fig. EV4H,I). One primary event in DDR is the rapid phosphorylation of histone H2AX by the phosphatidylinositol 3-kinase protein kinase-like family members, such as DNA-PK (DNA-dependent protein kinase) (Meyer et al, 2013). PRKDC is a catalytic subunit of DNA-PK which functions in DNA double-strand break repair and recombination (Blackford and Jackson, 2017). PKMYT1 deletion or inhibition enhanced Ser2056 phosphorylation of PRKDC in PDAC

cell lines and primary cultured cell (Figs. 4B and 6C), indicating PRKDC activation in human cells. Considering the DDR function of PRKDC, we speculate that PRKDC participates in DNA damage accumulation when PKMYT1 is inhibited. Consistently, NU7026 and NU7441, two PRKDC inhibitors attenuated γH2AX accumulation induced by the PKMYT1 inhibitor (Figs. 7A and EV5A; Dataset EV4), showing that PRKDC is involved in DDR induced by PKMYT1 inhibition. Then, we next sought to determine whether PRKDC participates in PKMYT1 inhibition-induced cytotoxicity. To this end, we treated PDAC cells with a wide range of NU7441 (or NU7026)/RP-6306 combinations. Isobologram and combination index (CI) analyses revealed that the combined treatment antagonistically inhibited cell growth with a CI > 1 for most concentration pairings (Figs. 7B,C and EV5B). *PRKDC* is amplified weakly in PDAC cell lines; thus RNAi but not CRISPR (Munoz et al, 2016) was used to interfere with its expression (Dataset EV2). *PRKDC* knockdown using shRNAs conferred resistance to RP-6306 on PDAC cells (Figs. 7D and EV5C). All these findings show PRKDC inhibition increases effective inhibitory concentration of RP-6306 and suppresses RP-6306-induced cytotoxicity and cell death in pancreatic cancer cells. In other words, activation of PRKDC is critical for the application of PKMYT1 inhibition for treatment in PDAC (Appendix Fig. S4).

## p53 activity participates in the dependence of PDAC on PKMYT1

There are two checkpoint arrests in the cell cycle that repair DNA damage to maintain genomic stability. Mutations of *TP53*, an important G1 checkpoint regulator, have been involved in the majority of human cancers (Aubrey et al, 2018). In total, 50–70% of human PDACs harbor *TP53* mutations, which occur at later stages of pancreatic intraepithelial neoplasia (PanIN; the precursor ductal lesion of PDAC), contributing to the malignant progression of PDAC (Kim et al, 2021). Mutations in the p53 pathways always lead to defective G1 checkpoint in many cancer cells (Matheson et al, 2016), resulting in increased DNA damage at the G2 checkpoint compared with normal cells (Chen et al, 2012). Abrogation of the G2 checkpoint forces cells with unrepaired DNA damage to enter mitosis prematurely, resulting in mitotic catastrophe and death. Cells with intact p53 signaling are less sensitive to G2 checkpoint abrogation as intact G1 checkpoint mechanisms cause a compensatory (Hamer et al, 2011). Therefore, G2 checkpoint abrogation is a promising strategy to preferentially damage cancerous cells relative to normal cells. Based on this, the correlations between *PKMYT1* expression and p53 pathway expressions were evident (Fig. 8A,B).

We established a mouse tumor cell line, namely, KC cells, which were derived from murine autochthonous pancreatic adenocarcinoma

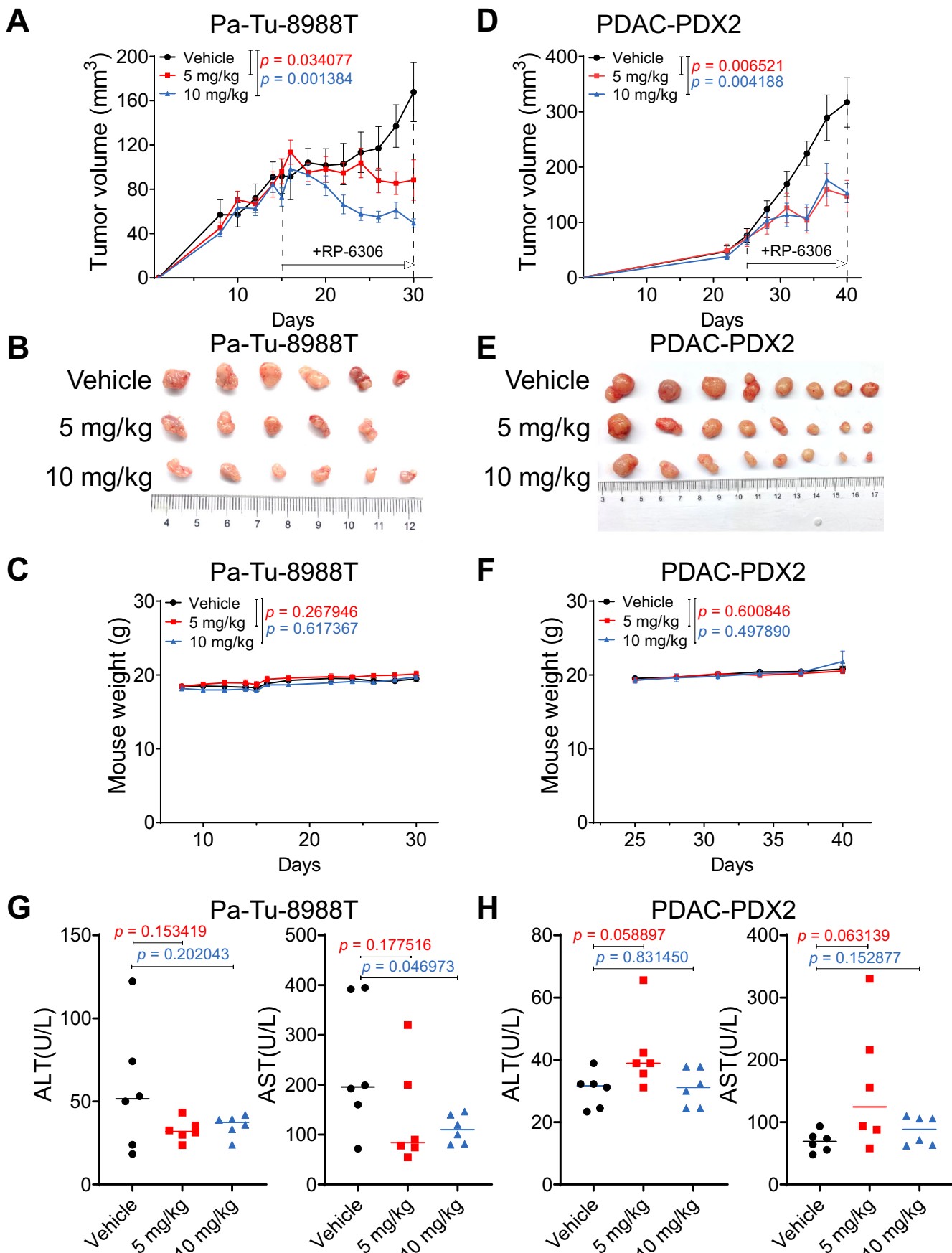

◄ **Figure 5.  RP-6306 suppresses tumor growth and proliferation in PDAC cell line xenografts and PDX models in vivo without overt toxicity.**

(A–F) Growth of Pa-Tu-8988T xenografts (A–C) and PDX (D–F) in BALB/c nude mice treated with either RP-6306 or vehicle. RP-6306 was administered orally twice daily at the indicated doses for the duration of the experiment. Growth curves (A, D), representative photo images (B, E) and mouse weight (C, F) of transplanted tumors are shown. The error bars indicate the mean ± s.d. of six replicates for Pa-Tu-8988T ($n = 6$ per group) or eight replicates for PDX ($n = 8$ per group); unpaired $t$ test. (G, H) RP-6306 treatment does not induce liver damage in nude mice. Alanine aminotransferase and aspartate aminotransferase activities in Pa-Tu-8988T xenografts (G) and PDXs (H) are shown. The error bars indicate the mean ± s.d. of six replicates ($n = 6$ per group); unpaired $t$ test. Source data are available online for this figure.

samples from $Pdx1^{cre}$; LSL-$Kras^{G12D}$ C57BL/6 mice. We used CRISPR/Cas9 to interrupt the $Trp53$ gene locus, and monoclonal sublines were established (Appendix Fig. S5A). According to the viewpoint mentioned above, $Pkmyt1$ should have little or no phenotypic effect on $Trp53$ intact subclones but should have profound effects on $Trp53$-mutant KC (KPC) cells. Interestingly, $Trp53$-mutant KC clonal cell lines (KPC#3, 10, 13) appeared to be more sensitive to the $Pkmyt1$-targeting sgRNA than KC cells (Fig. 8C; Appendix Fig. S5B). In addition, the $IC_{50}$ of RP-6306 in KPC cells was lower than that in KC cell line (Fig. 8D). In the long-term colony formation assay, $Pkmyt1$ ablation specifically suppressed the proliferation of $Trp53$-deficient cells (Fig. 8E). We also verified this finding in the human fibroblast cell line BJ and the GIST cell line with the wild-type $TP53$ gene (Fig. 8F), while the human PDAC cell lines used above contain $TP53$ mutations. For the sake of better elaborating this finding, we performed a subcutaneous tumor formation assay in nude mice. Knockout of $Pkmyt1$ significantly reduced the tumor growth rate of the KPC but not KC xenografts in nude mice (Fig. 8G–I). These data demonstrate that that p53 activity is involved in the essential function of PKMYT1 in PDAC (Appendix Fig. S4).

## Discussion

PDAC is a devastating disease, and therapeutic strategies used to treat PDAC in the clinic have achieved very limited benefit. In this study, we aimed to uncover genes required for PDAC proliferation to reveal novel targets for developing effective anticancer agents. To achieve this, we conducted systematic CRISPR/Cas9-based genome-wide functional screens to identify candidate essential genes involved in cell growth in PDAC. $PKMYT1$ is one of the top candidate hits. We provide evidence that PKMYT1 plays an oncogenic role in PDAC tumorigenesis and demonstrate that pharmacological inhibition is a very effective therapy against PDAC, both in human PDAC cell models and in human PDX models. Of equal relevance for future application of these evidences in a clinical scenario is the observation that systemic elimination of the PKMYT1 via pharmacological inhibition results in tolerable toxicities.

One of the most fundamental traits of cancer cells involves their ability to sustain proliferation (Hanahan, 2022). In more than 90% of PDAC cases, the key oncogenic driver is an activating $KRAS$ mutation (Buscail et al, 2020). It occurs as an early molecular event driving PanIN formation. Our functional studies indicated that PKMYT1 was highly expressed and essential in PDAC harboring oncogenic $KRAS$ mutations and in $KRAS$ wild-type PDAC, suggesting that PKMYT1 is a novel target in PDAC with a broad spectrum of molecular subtypes. Positive expression was found in a PDAC cohort consisting of 75 patients (36.0%) (Fig. 2A,B). $PKMYT1$ amplification is identified in 11% (12 of 109) of patients with PDAC in the UTSW cohort, consistent with the idea that

genomic amplification of the $PKMYT1$ locus in PDAC accounts for the high expression of PKMYT1. While gene amplification accounts for some of these high expression levels, it is not a widespread phenomenon. Studies have revealed that the down-regulation of the demethylase $ALKBH5$ leads to increased expression of $PKMYT1$ in gastric cancer (Hu et al, 2022). Further exploration into the epigenetic modifications of the $PKMYT1$ gene could provide insights into the mechanisms underlying its elevated expression, going beyond genomic amplifications.

Our findings first demonstrate that PKMYT1 plays a role in maintaining the stability of PLK1 and phosphorylates PLK1. Therefore, inhibiting PKMYT1 not only leads to suppression of tumor growth by regulating its own expression, but also exhibits an additional unexpected inhibitory effect through PLK1 regulation (Caron et al, 2016; Macurek et al, 2008). This is noteworthy considering the strong positive correlation between high $PLK1$ expression and tumor occurrence (Degenhardt and Lampkin, 2010).

Studies have shown that loss of either kinase interferes with the G2/M checkpoint, driving cells into mitosis prematurely (Chow and Poon, 2013). Loss of PKMYT1 dramatically influences the mitotic index of glioblastoma and human neural progenitors (Toledo et al, 2015), and a similar phenotype is observed in HeLa cells knockdown of PKMYT1 by siRNA (Villeneuve et al, 2013). Many of the $CCNE1$-high cells could skip G2 in response to PKMYT1 inhibition and do not go through a normal cell division but rather toggled between mitotic and interphase before terminating with high pan-$\gamma$H2AX signal (Gallo et al, 2022). $PKMYT1$ knockout or RP-6306 treatment inhibits cell cycle progression (Figs. 3H, 4D, and EV2F) and induces pan-$\gamma$H2AX in PDAC cancer cell line (Figs. 3I, 4C, and EV2C), which indicate that G2/M checkpoint dysfunction caused by PKMYT1 inhibition leads to unchecked premature mitotic entry, then results in the accumulation of genetic lesions from unrepaired DNA damage, ultimately leading to apoptosis or mitotic catastrophe (Figs. 3J, 4E and EV2G) (Asquith et al, 2020; Gallo et al, 2022). Cell cycle dysregulation, accompanied by aberrant activation of characteristic proteins, is a hallmark of cancer (Hanahan, 2022). Significant advancements have been made in the development of cancer therapies targeting cell cycle proteins, such as CDK4/6, resulting in highly effective and precise treatments for breast cancer (Li et al, 2022). $CCNE1$ amplification is synthetic lethal with PKMYT1 inhibition (Gallo et al, 2022). Pharmacological inhibitors of PKMYT1 have been designed and subsequently validated in $CCNE1$-amplified cancer models (Asquith et al, 2020; Gallo et al, 2022). Our data has shed further light on strategies for maximizing the response to PKMYT1 inhibition: (1) The activation of PRKDC plays a crucial role in modulating the sensitivity of PDAC to PKMYT1 inhibition; (2) PDACs with loss of $TP53$ function increase sensitivity to PKMYT1 inhibition. Further studies may

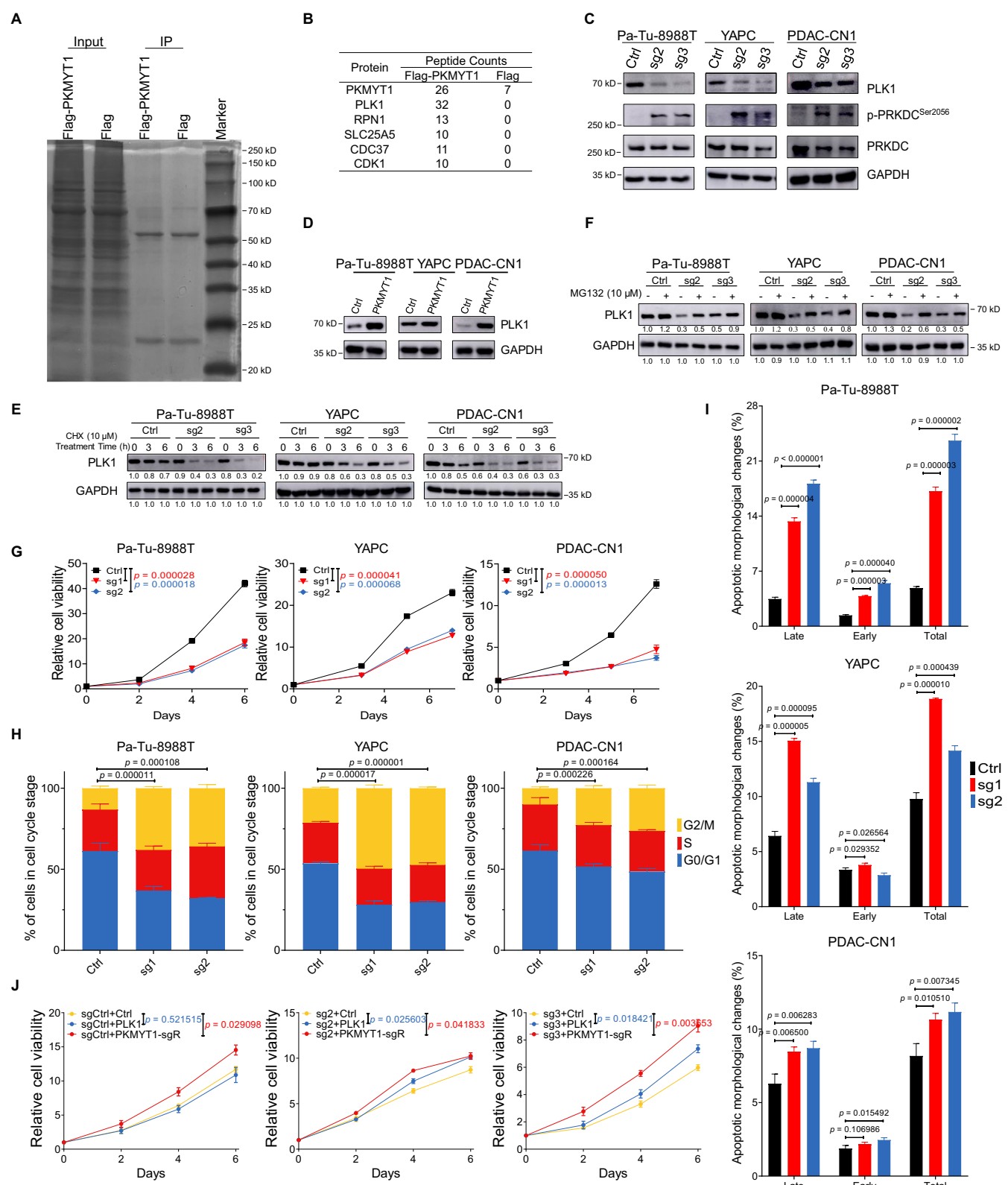

help patient stratification and maximize the response to PKMYT1 inhibition in clinical applications.

In summary, *PKMYT1* is highly expressed and serves as a potentially selective target in PDAC. PKMYT1 kinase interacts with CDK1 and PLK1 to function as an oncogene to promote PDAC tumorigenesis. The PKMYT1 inhibitor RP-6306 shows antitumor efficacy in human PDAC primary cell models and in human PDX models. Together, all the relevant experimental evidences support an

**Figure 6.** **PKMYT1 interacts with PLK1 and PLK1 mediates PKMYT1 oncogenic roles in PDAC.**

(A, B) Identification of PKMYT1-interacting proteins through co-immunoprecipitation followed by mass spectrometry-based proteomics analysis shows the most top abundant binding proteins. Pa-Tu-8988T cells were transduced with Flag-PKMYT1 or Flag lentivirus, and a pull-down assay was performed using anti-Flag antibody. Pull-down samples were detected by SDS-PAGE and followed by Coomassie staining (A). The table shows high-confidence hits in the mass spectrometry-based proteomics analysis (B). (C) Knockout of *PKMYT1* (sg2 and sg3) decreases PLK1 expression and increases PRKDC Ser2056 phosphorylation as indicated by western blotting. (D) PKMYT1 overexpression increases the protein level of PLK1, as indicated by western blotting. (E) Cells transduced with lentiviral vectors expressing sgCtrl and sgRNA targeting *PKMYT1* were treated with CHX (10 μM) for 0, 3, and 6 h. Knockout of PKMYT1 shortens the half-life of PLK1 as indicated by western blotting. (F) Cells transduced with lentiviral vectors expressing sgCtrl and sgRNA targeting PKMYT1 were treated with MG132 (10 μM) for 4 h. Knockout of *PKMYT1* inhibits the proteasome-mediated PLK1 degradation pathway, as indicated by western blotting. (G) *PLK1* knockout (sg1 and sg2) reduces cell viability in multiple PDAC lines and primary cultured cell PDAC-CN1, as assessed by a CellTiter-Glo viability assay. The error bars indicate the mean ± s.d. of three replicates; unpaired *t* test. (H) Knockout of *PLK1* causes G2/M arrest, as determined by flow cytometry. The error bars indicate the mean ± s.d. of three replicates; unpaired *t* test. (I) Flow cytometric analyses demonstrating increased apoptosis levels in PLK1-deleted Pa-Tu-8988T, YAPC and PDAC-CN1 cells. The error bars indicate the mean ± s.d. of three replicates; unpaired *t* test. (J) CellTiter-Glo viability assay of the indicated YAPC cells transduced with lentiviral vectors expressing sgRNA targeting *PKMYT1* or control sgRNA along with sgRNA-resistant PKMYT1-sgR, PLK1 or GFP. The proliferation inhibition is significantly attenuated by enforced PLK1 expression. The error bars indicate the mean ± s.d. of three replicates; unpaired *t* test. Ctrl control. (G, H) Represent data from two biological replicates. Source data are available online for this figure.

individualized treatment option for a PKMYT1-aberrant subgroup of PDAC patients receiving PKMYT1 inhibitor treatment.

# Methods

## Genes expression and patient survival analysis

To study the possible relationship between *PKMYT1* expression and survival, we first sorted patients from the TCGA-PAAD (https://www.cancer.gov/tcga) and GEO datasets into PKMYT1-high and PKMYT1-low groups according to the median of PKMYT1 transcriptional levels (top 50% versus bottom 50%). Next, we analyzed the difference in survival between these two groups using GraphPad Prism 9. TCGA-PAAD cohort along with GTEx data (https://www.cancer.gov/tcga) were used to compare differences in expression for indicated genes using the online database Gene Expression Profiling Interactive Analysis (GEPIA) (http://gepia.cancer-pku.cn/index.html) (Tang et al, 2017) or R (v4.2.2). The University of ALabama at Birmingham CANcer data analysis Portal (UALCAN) (http://ualcan.path.uab.edu/index.html) (Chandrashekar et al, 2022) was used to analyze the expression alteration of indicated genes based on different tumor grades and expression correlation between genes in TCGA-PAAD dataset. The Person and Spearman methods were used to determine the correlation coefficient.

## Cell lines and cell culture

HEK293T cells (American Type Culture Collection, ATCC #ACS-4500) and the PDAC cell lines Pa-Tu-8988T (Deutsche Sammlung von Mikroorganismen und Zellkulturen, DSMZ #ACC-162), YAPC (DSMZ #ACC-382), MiaPaCa-2 (ATCC #CRL-1420), BxPC-3 (ATCC #CRL-1687), SNU-324 (Courtesy Harvard Medical School Dr. Li Chen), KP-4 (Riken #RCB1005), Capan-1 (ATCC #HTB-79), Pa-Tu-8902 (DSMZ #ACC-179), DAN-G (DSMZ #ACC-249), and Panc.03.27 (ATCC #CRL-2549) were used. HEK293T, YAPC, Capan-1, Panc.03.27, DAN-G, BxPC-3 and SNU-324 cells were maintained in RPMI 1640 (HyClone #SH30027.01) medium containing 10% FBS (Thermo Fisher Scientific #10099141) and 1% penicillin/streptomycin (Thermo Fisher Scientific #15140122). Pa-Tu-8988T, MiaPaCa-2, Pa-Tu-8902, KP-4 and primary cells were maintained in DMEM/F12 (HyClone #SH30023.01) medium containing 10% FBS and 1% penicillin/streptomycin. All these cells were cultured at 37 °C in a 5% $CO_2$ humidified atmosphere. None

of the cell lines in this study appears in the misidentifed cell line list kept by the International Cell Line Authentication Committee. All cell lines were routinely tested for microbial contamination (including mycoplasma) and authenticated with transcriptome sequencing assays and sanger sequencing assays.

## Primary tumor cell culture

We established a mouse tumor cell line, namely, KC cells, which were derived from murine autochthonous pancreatic adenocarcinoma samples from *Pdx1^cre^*; LSL-*Kras^G12D^* C57BL/6 mice. Primary cultured cell (PDAC-CN1) was established from tissues obtained from surgical resection of the PDAC patient tumor samples. Tumor tissues were collected in serum-free DMEM/F12 medium and cut into small pieces (~2 mm in diameter), then cut into five small fragments not visible to the naked eye with a sterile eye scissors. Add collagenase type I (Gibco #17018029) to 100 U/mL with 3 mM $CaCl_2$ and incubate at 37 °C for 4 h on the shaker. Disperse cells by passing through a cell strainer. Wash dispersed cells several times by centrifugation in PBS. Seed cells into culture dish containing DMEM/F12 media. After 4 h, change fresh media. Since then, the cells were cultured in DMEM/F12 containing 10% fetal bovine serum and 1% penicillin/streptomycin mixed solution.

## Lentiviral gRNA library essentiality screens

Pa-Tu-8988T and YAPC were transduced with the GeCKOv2 library at a low multiplicity of infection (about 0.3). After infection cells experienced a 7-day puromycin (Millipore #540411) selection, then they were split into two independent replicate populations of minimum 300-fold library coverage (~37 million cells). A population-doubling 0 (PD0) sample was collected for genomic DNA extraction. The other replicate population was passaged without selection while maintaining a library fold-coverage of 300× for an additional 14 doublings (PD14). Genomic DNA was purified from PD0 and PD14 cell pellets, and guide sequence PCR was amplified with sufficient gDNA to maintain representation and quantified using deep sequencing.

## Screen data processing and analysis

Sequencing reads were processed by counting the number of unique reads for each sgRNA in each experimental group. sgRNAs with less than 50 counts in the PD0 control sample were removed

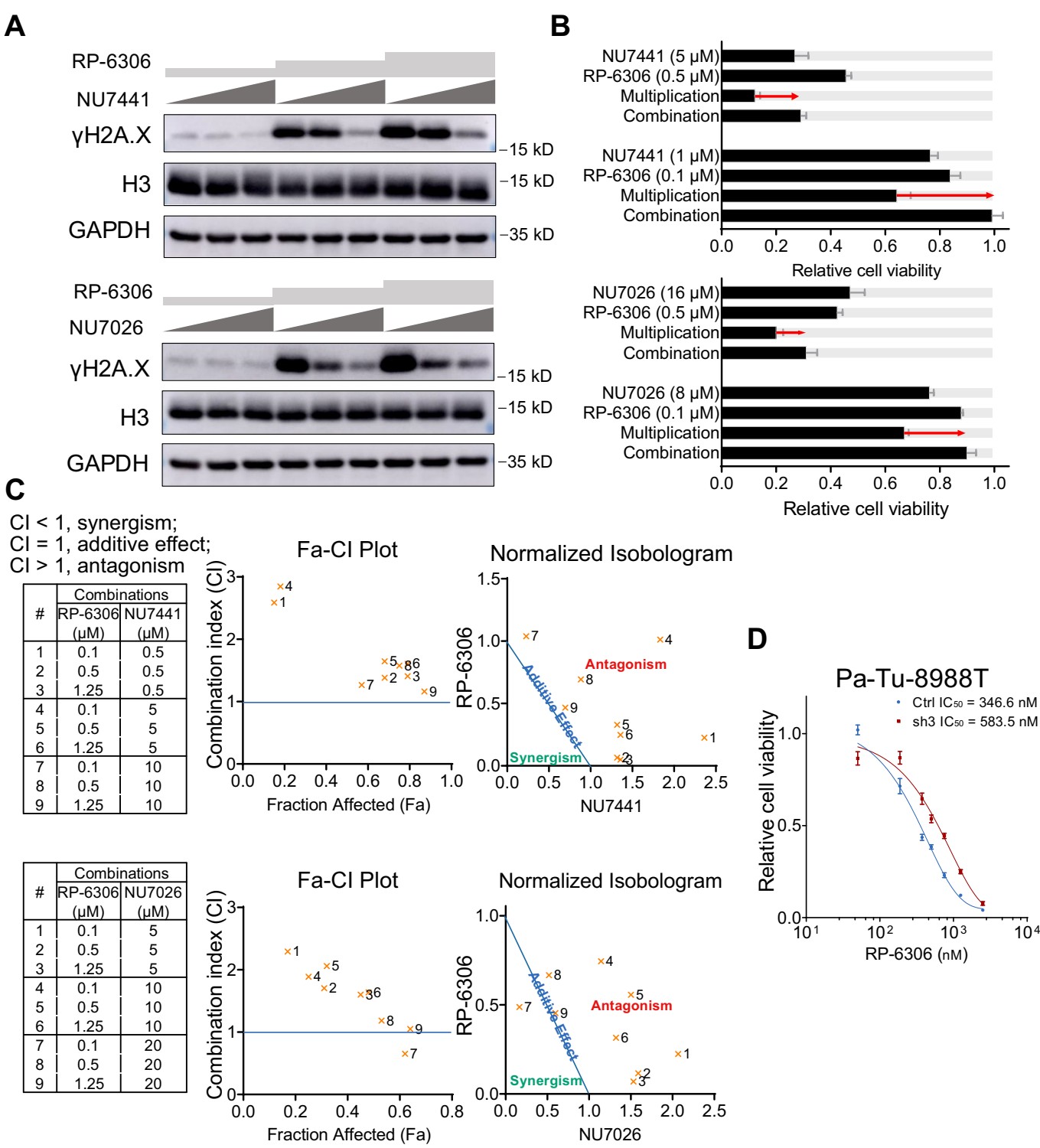

from downstream analyses. The log2-fold change in abundance of each sgRNA was calculated for final population samples for each of the cell lines after adding a count of one as a pseudocount, and the median of nontargeting controls in the GeCKOv2 library was subtracted from each sgRNA to generate a sgRNA score. Gene-based CRISPR scores (CS) were defined, and the the median score (med.CS) of all sgRNAs targeting a given gene is acted as a representative. MAGeCK-MLE algorithm was used to calculate to

obtain a β-score for each gene, which can also be used to describe the importance of each gene in the screening process.

## Plasmid constructs and lentivirus production

The PKMYT1 expression plasmid was constructed by cloning the corresponding cDNAs into the pCDH-CMV-MCS-EF1-Puro (or Neo) lentiviral expression vector (System Biosciences #CD510B-1).

**Figure 7. PKMYT1 ablation activates PRKDC, whose activity modulates the sensitivity of PDAC to PKMYT1 inhibition.**

(A) The PRKDC inhibitor NU7441 and NU7026 disturb γH2A.X accumulation induced by the RP-6306 treatment in Pa-Tu-8988T. As the concentration of RP-6306 (lane 1, 4, and 7) increases, γH2AX accumulation increases. For each dosage of RP-6306 (lanes 4–6, and lanes 7–9), γH2AX accumulation decreases in a NU7441/NU7026 concentration-dependent manner, showing that PRKDC is involved in DNA damage response induced by PKMYT1 inhibition. Concentrations of inhibitors are listed in "Methods". (B) Cell viability assay reveals the antagonistic effect between PRDKC inhibitors and RP-6306 in Pa-Tu-8988T. Light gray bars indicate control values. "Multiplication" indicates the expected effect of combined treatment if single-treatment effects are multiplied; red arrow indicates actual effect of the combination. The effect of PRDKC inhibitor and RP-6306 combined was even worse than would be expected if the individual effects were multiplied. The error bars indicate the mean ± s.d. of three replicates. (C) The PRKDC inhibitor NU7441 (left) and NU7026 (right) modulate the sensitivity of PDAC to PKMYT1 inhibitor in Pa-Tu-8988T. Combination index (CI) (top) and isobologram (bottom) analyses reveal the antagonistic effect between PKMYT1 inhibitor and PRKDC inhibitors in PDAC (CI > 1). Representative fraction affected (Fa)-CI plots (top) and normalized isobolograms (bottom) are shown ($n = 3$ per group). (D) *PRKDC* knockdown (sh3) desensitizes cells to RP-6306. $IC_{50}$ values are shown. The error bars indicate the mean ± s.e.m. of three replicates. Source data are available online for this figure.

3XFlag sequence was added just ahead the amino terminus of PKMYT1 to obtain the Flag-tag PKMYT1. CRISPR-resistance PKMYT1 expression plasmid was constructed by introducing synonymous mutations using Seamless Cloning Kit (Beyotime #D7010FT), and on this basis, PKMYT1–N238A point mutant constructed. The PLK1 expression plasmid was constructed by cloning the corresponding cDNAs into the pCDH-CMV-MCS-EF1- Puro (or Neo) lentiviral expression vector. The recombinant Myc-PLK1 (and Myc-CDK1) were expressed using a pCDH expression vector as a fusion protein with the C-terminal tag. The phospho-mimicking PLK1 mutants are constructed by introducing synonymous mutations using Seamless Cloning Kit. The short guide RNAs (sgRNAs) that target human *PKMYT1*, *PLK1*, *Trp53*, and *Pkmyt*1 were designed using the Optimized CRISPR Design web tool (http://crispr.mit.edu/). These sgRNA vectors were generated by cloning the guide sequences into the lentiCRISPRv2 vector (Addgene #52961) individually. The short hairpin RNAs (shRNAs) that target human *PRKDC* were designed using the GPP Web Portal (https://portals.broadinstitute.org/gpp/public/). *PRKDC* shRNA vectors were generated by cloning these sequences into the pLKO.1 vector (Sigma #SHC001) individually. Empty vectors were used as control. The primers and other sequences are listed in Dataset EV2. Lentivirus particles were generated by co-transfecting these constructs with helper virus packaging plasmids pCMVΔ8.9 and pHCMV-VSV-G into HEK293T cells using Lipofectamine 3000 (Invitrogen #L3000015) or polyethylenimine (Sigma #408727) for library package. Lentivirus were harvested after 24, 36, 48, and 60 h, and frozen at −80 °C in aliquots at appropriate amounts for infection. All cells were infected with lentivirus for 16 h in the supernatant containing 8 μg/mL polybrene (Sigma #107689), and then treated with puromycin (Millipore #540411) one day after infection.

## Soft-agar assay and colony formation assay

Six-well plates were first layered with 0.6% bottom agar (BD difco #214220) containing RPMI 1640 medium with 10% FBS and 1% penicillin/streptomycin. Pa-Tu-8988T (8000 per well), YAPC (25000 per well) transduced with PKMYT1 sgRNAs or Ctrl were seeded in 0.35% top agar containing 10% FBS and 1% penicillin/streptomycin. Cells were allowed to grow for 3–5 weeks and then stained with 1 mL of 1 mg/mL methyl thiazol tetrazolium (Sigma-Aldrich #M5655) for 4 h. Colonies area percent was measured by ImageJ software (National Institutes of Health). All assays were performed in triplicate wells, with the entire study replicated at least once. Colony formation assay was conducted by seeding Pa-Tu-8988T (250 per well), YAPC (1000 per well), PDAC-CN1 (500

per well), KC (750 per well), KPC#10 (750 per well) cells transduced with PKMYT1 sgRNAs or Ctrl into six-well plates and allowed to grow for 2–4 weeks. Then, the cells were fixed with 4% paraformaldehyde for 30 min and stained with crystal violet solution (Sangon Biotechnology #E607309-0100) for 30 min. All assays were carried out in triplicate wells, with the entire study replicated at least once. Images were obtained using a scanner (Microtek #1600III). Colonies were counted by ImageJ software.

## Inhibitors treatment

For in vitro experiments, inhibitors were dissolved with dimethyl sulfoxide (DMSO). According to the sensitivities for inhibitor treatment, cells were exposed to inhibitors in different concentration gradients. For PKMYT1 inhibitor RP-6306 (MCE #HY-145817A), a series of drug concentrations were applied for Pa-Tu-8988T (0, 0.1, 0.25, 0.75, and 2.5 μM), YAPC (0, 1, 2.5, 5, and 7.5 μM) and PDAC-CN1 (0, 0.05, 0.1, 0.25, 0.5, and 1 μM) when validating the regulation for PLK1, PRKDC and CDK1. Pa-Tu-8988T (0, 0.1, 0.25, and 0.5 μM), YAPC (0, 1, 2.5, and 5 μM) and PDAC-CN1 (0, 1, 2.5 and 5 μM) were treated for 24 h or 48 h with RP-6306 by indicated concentration to reflect the γH2A.X accumulation. Dasatinib was applied to Pa-Tu-8988T (0, 0.1, 0.5, and 1 μM), YAPC (0, 1, 5, and 10 μM) and PDAC-CN1 (0, 0.1, 0.5, and 1 μM). Cells and primary cultured cell were treated for 4 h or 12 h with PD0166285 (MCE #HY-13925; 0, 0.1, 0.5, and 1 μM). For PLK1 inhibitor onvansertib (Selleck #S7255), cells were treated for 48 h (Pa-Tu-8988T: 0, 0.05, 0.1 and 0.5 μM; YAPC: 0, 1, 5, and 10 μM). To evaluate the efficacy of RP-6306 in combination with PRKDC inhibition, each PRKDC inhibitor had three concentrations (NU7441: MCE #HY-11006; 0, 1, and 5 μM; NU7026: MCE #HY-11719; 0, 8, and 16 μM) to be combined with different PR-6306 dosages (Pa-Tu-8988T: 0, 0.25, and 1 μM; YAPC: 0, 2.5, and 5 μM; PDAC-CN1: 0, 1, and 5 μM). Gemcitabine (MCE #HY-17026) was used to test synergistic effect in Pa-Tu-8988T and YAPC. Other details for inhibitor treatment are indicated in the main text and figure legends.

## Cell viability assays, isobologram, and combination index analysis

Viability studies were performed using the CellTiter-Glo luminescent assay (Promega #G7573). Cells were plated at $0.75–3 \times 10^4$ cells per well in a 96-well flat-bottomed plate. To evaluate cell proliferative potential, viability assays were conducted at the indicated timepoints. PKMYT1, PRKDC and/or PLK1 inhibitors with indicated concentrations were added to to 96-well plates after

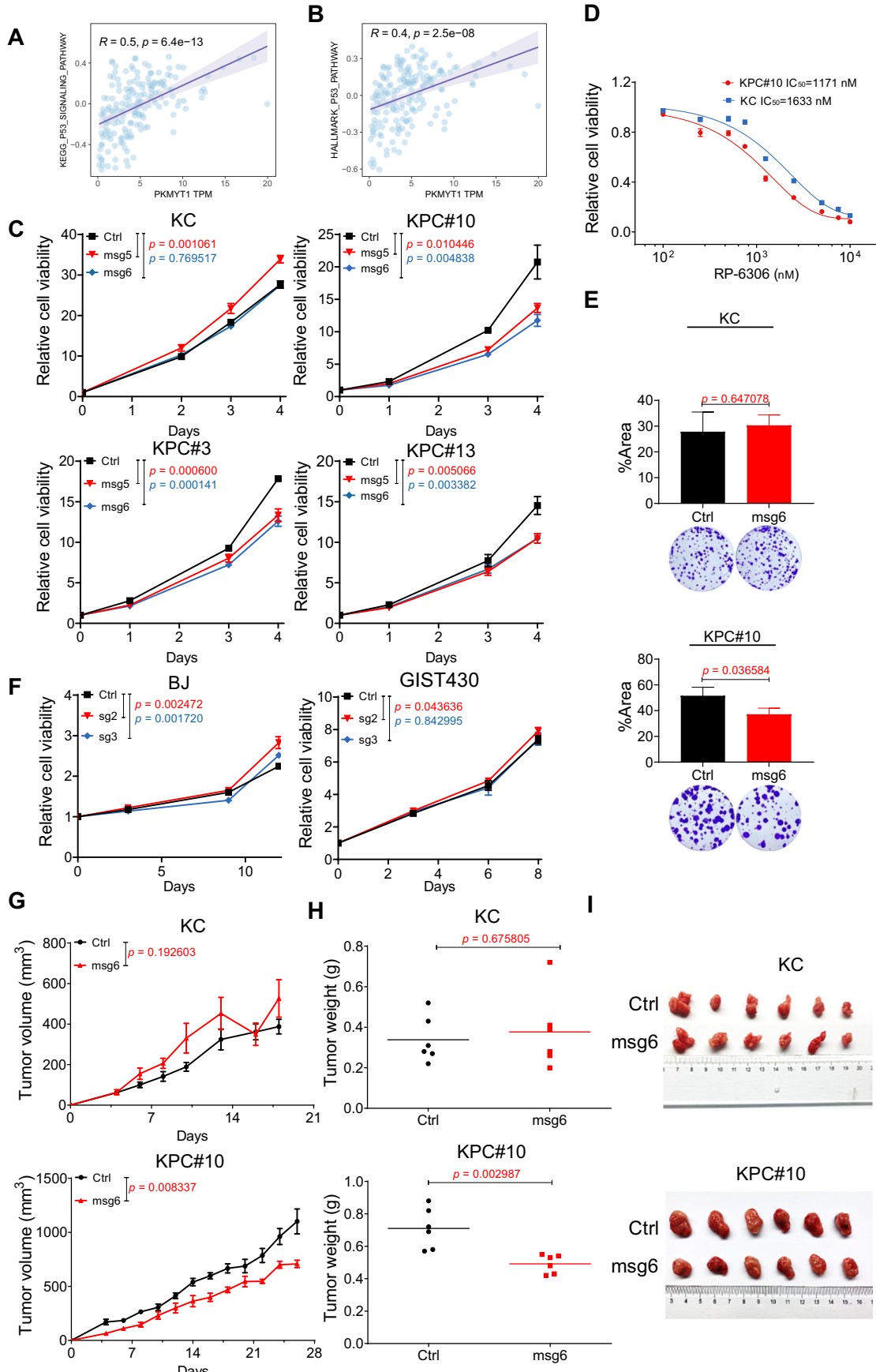

Figure 8. The function of p53 affects PKMYT1 dependency in PDAC.

(A, B) Positive correlation between the *PKMYT1* expression level and p53 pathway. Data were retrieved from the TCGA-PAAD dataset ($n = 179$). Unpaired *t* test; pearson correlation coefficient was used. (C) *Pkmyt1* depletion (msg5 and msg6) shows slight toxicity to cells with the wild-type *TP53* gene. *PKMYT1* knockout does not affect the viability of KC cells but specifically inhibits the viability of KPC cells, as assessed by a CellTiter-Glo viability assay. The error bars indicate the mean ± s.d. of three replicates; unpaired *t* test. (D) *Trp53* inactivation sensitizes PDAC cells to RP-6306. $IC_{50}$ values are shown. The error bars indicate the mean ± s.e.m. of three replicates. (E) Quantification and representative plates of crystal violet staining assays show that *Pkmyt1* ablation specifically suppresses the proliferation of *Trp53*-deficient cells. The error bars indicate the mean ± s.d. of three replicates; unpaired *t* test. (F) *PKMYT1* depletion (sg2 and sg3) shows slight toxicity to cells with the wild-type *TP53* gene, as assessed by a CellTiter-Glo viability assay. The error bars indicate the mean ± s.d. of three replicates; unpaired *t* test. (G-I) *Pkmyt1* deletion inhibits the growth of KPC xenografts but not KC xenografts in nude mice. Growth curves (G), tumor weight (H) and representative photo images (I) of transplanted tumors are shown. The error bars indicate the mean ± s.d. of six replicates ($n = 6$ per group); unpaired *t* test. (C, E, F) Represent data from two biological replicates. Source data are available online for this figure.

24 h. Viability studies were performed once every two or three days. Luminescence was analyzed using a BioTek Gen5 Microplate Readers (BioTek #H1210-018). Isobologram analysis was performed using CalcuSyn software, version 2 (Biosoft, Cambridge, UK). The fraction affected (Fa) was calculated from cell viability assays, and the combination index (CI) was generated using CalcuSyn software (Biosoft, Cambridge, UK) as described previously (Chou, 2010; Gabrielsson et al, 2016).

## Western blotting, immunoprecipitation, and mass spectrometry

Whole-cell lysates from cell lines were prepared using IP buffer (50 mM Tris-HCl, pH 8.0, 1% NP-40, 100 mM sodium fluoride, 2 mM sodium molybdate, 30 mM sodium pyrophosphate, 5 mM EDTA, and 2 mM sodium orthovanadate) containing protease inhibitors (10 µg/mL leupeptin, 10 µg/mL aprotinin, 1 mM phenylmethylsulfonyl fluoride and 10 µg/mL vanadate). The lysates were then rocked for 8 h or crushed by ultrasonic Cell Disruptor (40% Amp, 16 s) at 4 °C and cleared by centrifugation at 14,000 rpm for 30 min at 4 °C. The protein concentrations in the lysate were determined using a Quick Start Bradford 1× Dye Reagent (Bio-Rad #5000205).

Electrophoresis and western blotting were performed using standard techniques. The hybridization signals were detected by chemiluminescence (GE Healthcare #RPN2134) and captured using an Amersham Imager 600 imagers (GE Healthcare #29083461). The primary antibodies were list as follows: PKMYT1 (Cell Signaling Technology #4282S, 1:1000; Santa Cruz #sc-74523, 1:500), CDK1 (Santa Cruz #sc-54, 1:200), Phospho-CDK1 (Thr14) (Abclonal #AP0016, 1:1000), Histone H3 (Cell Signaling Technology #12648S, 1:1000), Phospho-Histone H2A.X (Ser139) (Cell Signaling Technology #9718S, 1:1000), PCNA (Santa Cruz #sc-56, 1:750), β-actin (Abclonal #AC026 1:50000), GAPDH (ABMART #M20006 1:5000), PLK1 (Abclonal #A21082 1:1000; ABMART #MG670393, 1:1000), PRKDC (Abclonal #A1419, 1:1000), Phospho-PRKDC (Ser2056) (Abclonal #AP0621, 1:1000), Flag-Tag (Merck #F1804, 1:1000), Myc-Tag (Abclonal #AE070, 1:10000; Santa Cruz #sc-40, 1:750), Pan-phospho-Ser/Thr (Abclonal #AP0893, 1:1000), Phospho-PLK1-Thr210 (Abclonal #AP1025, 1:1000) and Phospho-PLK1–Tyr217 (ABMART #TA7322, 1:1000). Relative protein quantification was performed by ImageJ software.

For immunoprecipitation, 2.5 mg whole-cell lysates were mixed with 1 µg anti-Flag-Tag (Merck #F1804) antibody and 20 µL protein G-sepharose (Invitrogen #10-1243), incubated overnight, then washed by IP buffer and eluted by boiling with SDS loading buffer.

3-5 mg whole-cell lysates were mixed with 2 µg anti-PKMYT1 (Santa Cruz #sc-74523), anti-PLK1 (ABMART #MG670393) or IgG (Santa Cruz #sc-2025; Abclonal #AC005) antibody and 40 µL protein G-sepharose (Thermo Fisher Scientific #101242), incubated overnight, then washed by IP buffer and eluted by boiling with SDS loading buffer. The eluted samples were detected by SDS-PAGE followed by Coomassie staining (Invitrogen #LC6025). For mass spectrometry, IP samples were eluted by shaking with 8 M urea and 100 mM Tris-Cl, pH 8.0, and analyzed by mass spectrometry. IP samples and whole-cell lysates were analyzed by western blotting.

## In vitro kinase assay

Immunoprecipitates of recombinant Flag-PKMYT1 (~7 mg overexpression-whole-cell lysates incubated with Anti-Flag Affinity Gel (Selleck #B23101) at 4 °C overnight, followed by 3 washes with PBST buffer then eluted by 500 µg/mL 3× Flag Peptide (Beytime #P9801)) and recombinant Myc-PLK1 or Myc-CDK1 (0.5 mg overexpression-whole-cell lysates incubate with rec-Protein G-Sepharose (Invitrogen #10-1243) and anti-Myc-Tag antibody at 4 °C overnight, followed by three washes with IP buffer and one wash with 10 mM Tris-HCl pH 7.4). Myc-PLK1 was resuspended in 30 µL kinase buffer (Cell Signaling Technology #9802) containing 20 µL recombinant Flag-PKMYT1, 200 µM ATP (Beytime #D7378), phosphatase inhibitor cocktail (Beytime #D7378) and 1 µM RP-6306 or vehicle (MCE #HY-145817A). After incubation for 1 h at 37 °C, reactions were stopped on ice by the addition of 2.5× protein loading buffer and boiling for 5 min at 95 °C. Samples were separated on SDS-PAGE. Phosphorylation was detected by western blotting.

## Immunohistochemistry

Formalin-fixed paraffin-embedded surgical clinical samples were obtained from Shanghai Jiao Tong University School of Medicine Affiliated Renji Hospital and with institutional review board approval (KY2020-116). Immunohistochemistry was performed on tissue microarrays using PKMYT1 antibody. In brief, the slides were dehydrated through a 20%, 80%, 95%, and 100% ethanol series, cleaned in xylene, then boiled by microwave for 12 min in citrate buffer (pH 6). Immunohistochemistry reactions were visualized by diaminobenzidine staining using an EnVision+ system (Dako).

## Immunofluorescence

Cells grown on slides were rinsed with phosphate-buffered saline (PBS), fixed with 4% paraformaldehyde for 15 min, and permeabilized

in 0.5% Triton X-100 in PBS. The slides were incubated with the primary antibody for 1 h and incubated for 1 h with the appropriate fluorescent-labeled secondary antibody. PKMYT1 (Santa Cruz #sc-74523, 1:50), PLK1 (ABMART #MG670393 1:100), Phospho-Histone H2A.X (Ser139) (Cell Signaling Technology #9718S, 1:200).

## Cell cycle and apoptosis assays

For cell cycle analysis, cells were selected by puromycin for 3 days, serum-starved for 24 h, and then cultured using complete medium for another 24 h. Cells were harvested and fixed in 75% ethanol for 24 h, resuspended in 50 µg/mL of propidium iodide (Sigma-Aldrich #P4170) and 100 µg/mL of RNaseA (TIANGEN #RT405) containing PBS solution after centrifugation, then were analyzed using the Gallios Flow Cytometer (Beckman Coulter) and FlowJo_V10 software. For apoptosis analysis, cells were cultured for 5 days after infection, then washed with PBS and harvested. Cells were stained with the APC Annexin V Apoptosis Detection Kit with 7-AAD (BioLegend #640930) and evaluated by flow cytometry (Beckman Coulter; Gallios Flow Cytometer) according to the manufacturer's protocol.

## Transcriptome sequencing

Total RNA was isolated from cells using the standard Trizol protocol. Paired-end sequencing ($2 \times 100$ bp) was performed with a BGI-500 instrument to obtain at least 20 million reads for each sample. The sequence data were processed and mapped to the human reference genome (hg19) using Bowtie2. Gene expression was quantified to fragments per kilobase per million mapped fragments using RNA sequencing by expectation maximization.

## Cell line-derived and patient-derived xenografts

Female BALB/c nude mice (5–6 weeks old; weight 18–25 g) were obtained from Shanghai Lingchang BioTech Co. Ltd, PR. China. The mouse studies were approved by the Institutional Animal Care and Use Committee (IACUC) of the Shanghai Institutes for Biological Sciences. All animals were maintained in the specific pathogen-free facility of Shanghai Institutes for Biological Sciences, Chinese Academy of Sciences, according to the international, national and institutional guidelines for humane animal treatment and complied with relevant legislation (SIBS-2019-WYX-1; SINH-2020-WYX-1; SINH-2021-WYX-1; SINH-2022-WYX-1). Pa-Tu-8988T ($2 \times 10^6$ cells), YAPC ($2 \times 10^6$ cells), PDAC-CN1 ($2 \times 10^6$ cells), KC ($2 \times 10^6$ cells) and KPC#10 ($1.75 \times 10^6$) cells transduced with PKMYT1 sgRNAs or Ctrl were injected into the right flanks of mice. Six mice in each group. Tumor xenografts were allowed to grow for 3–6 weeks. Once the largest tumor diameter closed to the maximal tumor diameter allowed under our institutional protocol or met observation criteria, all mice were sacrificed, and tumors were collected. The maximal tumor diameter allowed by the Institutional Animal Care and Use Committee was 2.0 cm. For inhibitor treatment in vivo, when tumors reached the target size of 100–150 mm³, mice were randomized to treatment group or vehicle group ($n = 6$ in each group) according to tumor volume and body weight, and treatment with RP-6306 was initiated. RP-6306 was formulated in 1.5% DMSO + 40% PEG300 + 5% Tween-80 + 53.5% saline, and orally administered twice daily (BID, 0–8 h) for a maximum of 15 days. The vehicle consists of the solvent described above.

For patient-derived xenografts, the study protocol was reviewed and approved by the Ethics Committee of Shanghai Jiao Tong University School of Medicine Affiliated Renji Hospital (KY2020-116). Informed written consent was obtained from all human participants. Fresh primary human tumor tissue was collected and cut into small pieces (~2 mm in diameter). These tumor fragments were inoculated subcutaneously into the right flank of nude mice for tumor development and subsequently passaged by implantation into the cohort of mice enrolled in the efficacy study. Mice were randomized according to growth rate into the treatment group or vehicle group ($n = 8$ in each group) when the mean tumor size reached ~150 (100–200) mm³. Animals were monitored for tumor volume, clinical signs and body weights. Tumor volume was measured using a digital caliper and calculated using the formula: tumor volume $= 0.5 \times L \times W \times W$, where L is length and W is width. After the last administration, fasting whole blood was collected 8 h later. After RP-6306 administration, blood was collected from 6 mice in each group for liver function and routine peripheral blood tests. Note that blood collection was not successful from two of the eight mice in the vehicle group (PDX). Therefore, blood from 6 mice in the RP-6306 groups were used for the liver function and blood tests. The mice with the largest and smallest volume in the RP-6306 groups were excluded for the liver function and blood tests. No animals were excluded from the other analyses.

## Human samples

For formalin-fixed paraffin-embedded surgical clinical samples (tissue microarray, TMA) and patient-derived xenografts were obtained from Renji Hospital. The study protocol was reviewed and approved by the Ethics Committee of Shanghai Jiao Tong University School of Medicine Affiliated Renji Hospital (KY2020-116). All samples were collected with institutional review board approval. Informed written consent was obtained from all human participants. All the experiments conformed to the principles set out in the WMA Declaration of Helsinki and the Department of Health and Human Services Belmont Report.

## Statistical analysis

Unpaired *t* test was used to compare the statistical differences for two conditions or groups if not mentioned specifically in the figure legends, and assume both samples in each condition or group are from populations with the same standard deviation. All experiments were indicated as the mean ± s.d. or s.e.m. of at least three independent biological replicates, The Specific number of sampled units are indicated in the figure legends. Female BALB/c nude mice were grouped before the subcutaneous tumor model was established at random. Mice were randomized according to growth rate into treatment group or vehicle group when the mean tumor size reached ~150 (100–200) mm³ for inhibitor treatment in vivo. No blinding was performed in the research. Statistical analysis was performed using GraphPad Prism 9 (GraphPad Software). IC$_{50}$ of inhibitor in each group was analyzed using nonlinear regression model.

**The paper explained**

**Problem**

Pancreatic ductal adenocarcinoma (PDAC) is a devastating disease with an overall 5-year survival rate of <12% due to the lack of effective treatments. Novel treatment strategies are urgently needed with limited therapeutic options and dismal long-term survival.

**Results**

In this paper, we performed whole-genome CRISPR/Cas9 loss-of-function screens and identified *PKMYT1* as a potential therapeutic candidate for PDAC. *PKMYT1* is frequently overexpressed and higher protein expression levels indicate poor prognosis in PDAC patients. PKMYT1 ablation inhibits tumor growth and proliferation in vitro and in vivo. Our data also shed further light on strategies for maximizing the response to PKMYT1 inhibition: (1) The activation of PRKDC; (2) PDACs with a loss of *TP53* function. Moreover, RP-6306, an orally bioavailable inhibitor, is found to effectively inhibit PKMYT1 selectively in PDAC.

**Impact**

These results define PKMYT1 dependency in PDAC and identify potential therapeutic strategies for clinical translation. This evidence supports a personalized therapeutic option for a novel molecular subgroup of patients with PKMYT1-aberrant PDAC receiving PKMYT1 inhibitor treatment.

# Data availability

RNA-Seq data: the National Omics Data Encyclopedia accession no. OEP004116 https://www.biosino.org/node/experiment/detail/OEX023088. CRISPR screens data: the National Omics Data Encyclopedia accession no. OEP004116 https://www.biosino.org/node/experiment/detail/OEX023089.

# Peer review information

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

## Acknowledgements

This work was supported by grants from the National Natural Science Foundation of China (82072974, 82120108020); the Basic Research Project of Shanghai Science and Technology Commission (20JC1419200); the Innovation Program of Shanghai Science and Technology Committee (20Z11900300); the National Key Research and Development Program of China (2023YFE0117900). We thank Zhonghui Weng and Kai Wang from the animal facility of Shanghai Institute of Nutrition and Health for animal care during the coronavirus pandemic.

## Author contributions

**Simin Wang**: Conceptualization; Resources; Data curation; Software; Validation; Investigation; Visualization; Methodology; Writing—original draft; Writing—review and editing. **Yangjie Xiong**: Software; Investigation; Methodology. **Yuxiang Luo**: Investigation; Methodology. **Yanying Shen**: Supervision; Investigation; Visualization; Methodology. **Fengrui Zhang**: Methodology. **Haoqi Lan**: Investigation; Methodology. **Yuzhi Pang**: Software; Investigation; Methodology. **Xiaofang Wang**: Investigation; Methodology. **Xiaoqi Li**: Investigation; Methodology. **Xufen Zheng**: Investigation; Methodology. **Xiaojing Lu**: Investigation; Methodology. **Xiaoxiao Liu**: Investigation; Methodology. **Yumei Cheng**: Investigation; Methodology. **Tanwen Wu**: Investigation; Methodology. **Yue Dong**: Investigation; Methodology. **Yuan Lu**: Supervision. **Jiujie Cui**: Supervision. **Xiaona Jia**: Data curation. **Sheng Yang**: Supervision; Methodology; Writing—review and editing.

**Liwei Wang**: Data curation; Supervision; Methodology; Writing—review and editing. **Yuexiang Wang**: Conceptualization; Data curation; Supervision; Funding acquisition; Visualization; Methodology; Writing—original draft; Writing—review and editing.

## Disclosure and competing interests statement

Yuexiang Wang, Simin Wang, and Xiaona Jia report a patent no. 2023108819852 pending. The remaining authors declare no competing interests.

# Expanded View Figures

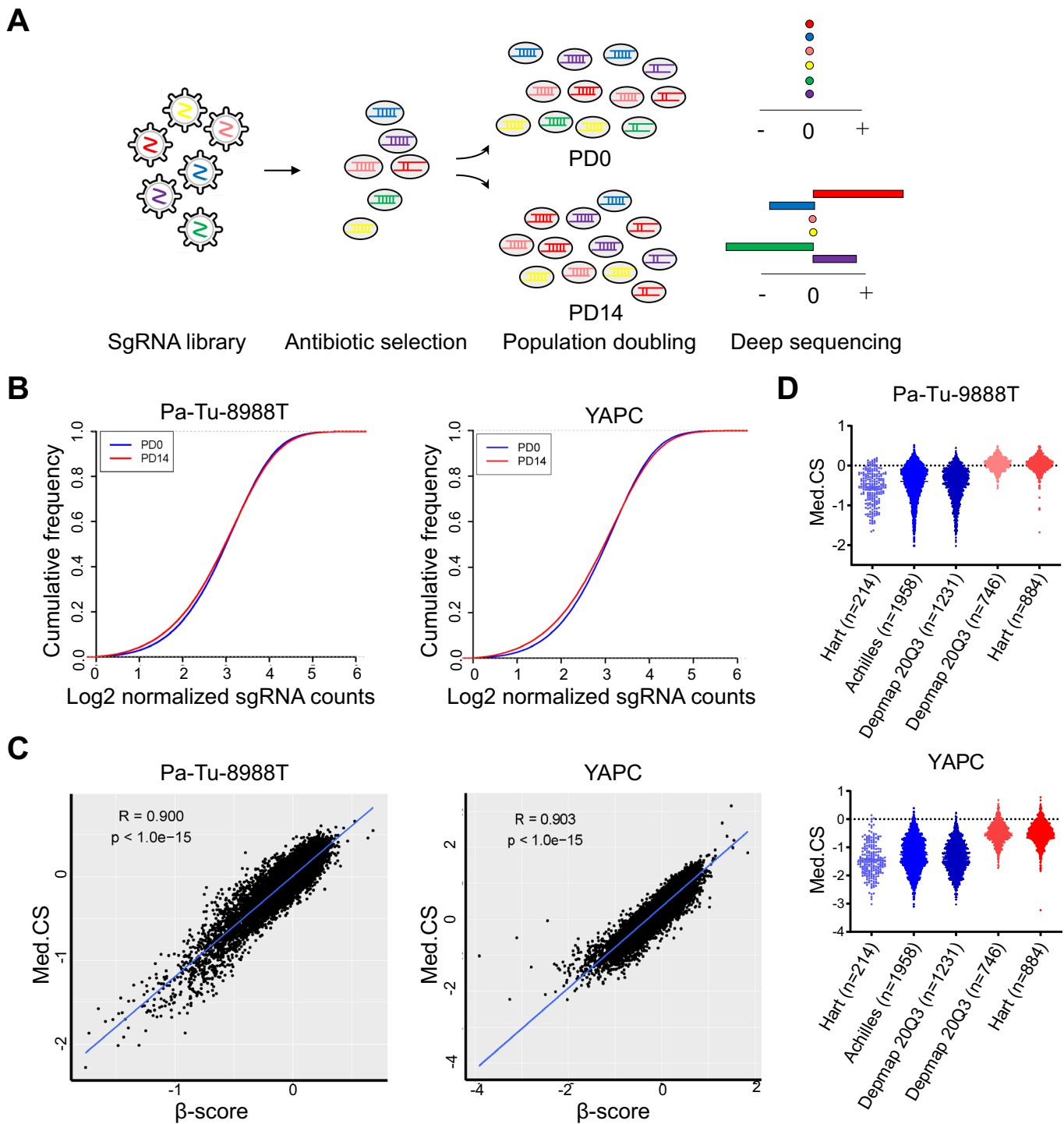

**Figure EV1.   Genome-wide CRISPR screen overview.**

(**A**) Schematic outlining the design of the genome-wide CRISPR/Cas9 knockout screens in PDAC cells. (**B**) Cumulative frequency of sgRNAs during the 14 PDs after transduction in Pa-Tu-8988T and YAPC cells. The shift in the PD14 curve indicates the depletion of a subset of sgRNAs. (**C**) Significant correlation between the med.CS and β score in the Pa-Tu-8988T and YAPC cell lines. Unpaired *t* test; pearson correlation coefficient was used. (**D**) Plot comparing the med.CS of known pan-essential (blue points) and non-pan-essential genes (red points) in Pa-Tu-8988T and YAPC cells. *n* indicates the number of genes.

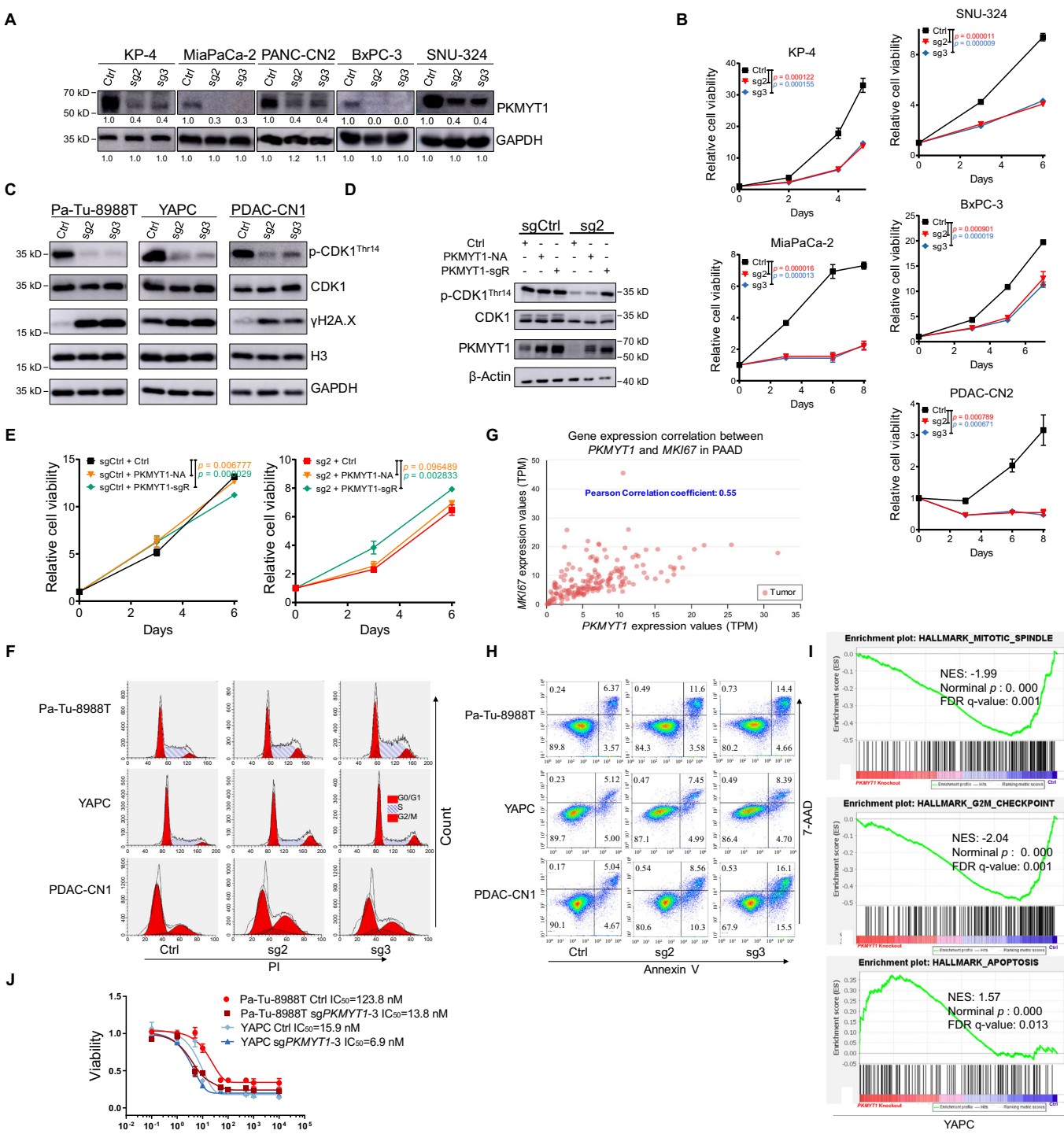

◀ **Figure EV2. Depletion of *PKMYT1* inhibits tumor growth and proliferation in PDAC models in vitro and in vivo.**

(A) CRISPR-mediated *PKMYT1* knockout (sg2 and sg3) decreases PKMYT1 protein expression in four more cell lines (KP, Miapaca2, SNU-324 and Bxpc3) and one more primary cultured cell (PDAC-CN2). (B) Lentivirus-mediated *PKMYT1* knockout reduces the viability of cells in Fig. EV2A. The error bars indicate the mean ± s.d. of three replicates; unpaired *t* test. (C) Knockout of *PKMYT1* decreases CDK1 Thr14 phosphorylation and increases γH2A.X accumulation. (D, E) CellTiter-Glo viability assay of the indicated YAPC cells transduced with lentiviral vectors expressing sgRNA targeting *PKMYT1* (sg2) or control sgRNA (sgCtrl) along with sgRNA-resistant PKMYT1 (PKMYT1-sgR) or kinase-dead sgRNA-resistant PKMYT1-N238A (PKMYT1-NA). The protein kinase activity of PKMYT1 is, at least partially, required for the regulation of tumor proliferation. The error bars indicate the mean ± s.d. of three replicates; unpaired *t* test. (F) Representative results of cell cycle analysis by flow cytometry for Fig. 3H. (G) *PKMYT1* expression correlates with proliferative index indicated by Ki67 ($n = 179$). (H) Representative results of apoptosis analysis by flow cytometry for Fig. 3J. (I) GSEA of differentially expressed genes in YAPC cells demonstrates that *PKMYT1* knockout regulates the genes involved in the cell cycle (including mitotic spindle, and G2M checkpoint) and apoptosis pathway. NES normalized enrichment score. (J) *PKMYT1* depletion sensitizes pancreatic cancer cells to gemcitabine. The error bars indicate the mean ± s.d. of three replicates. Figures EV2C, EV2F, and EV2H represent data from two biological replicates. Source data are available online for this figure.

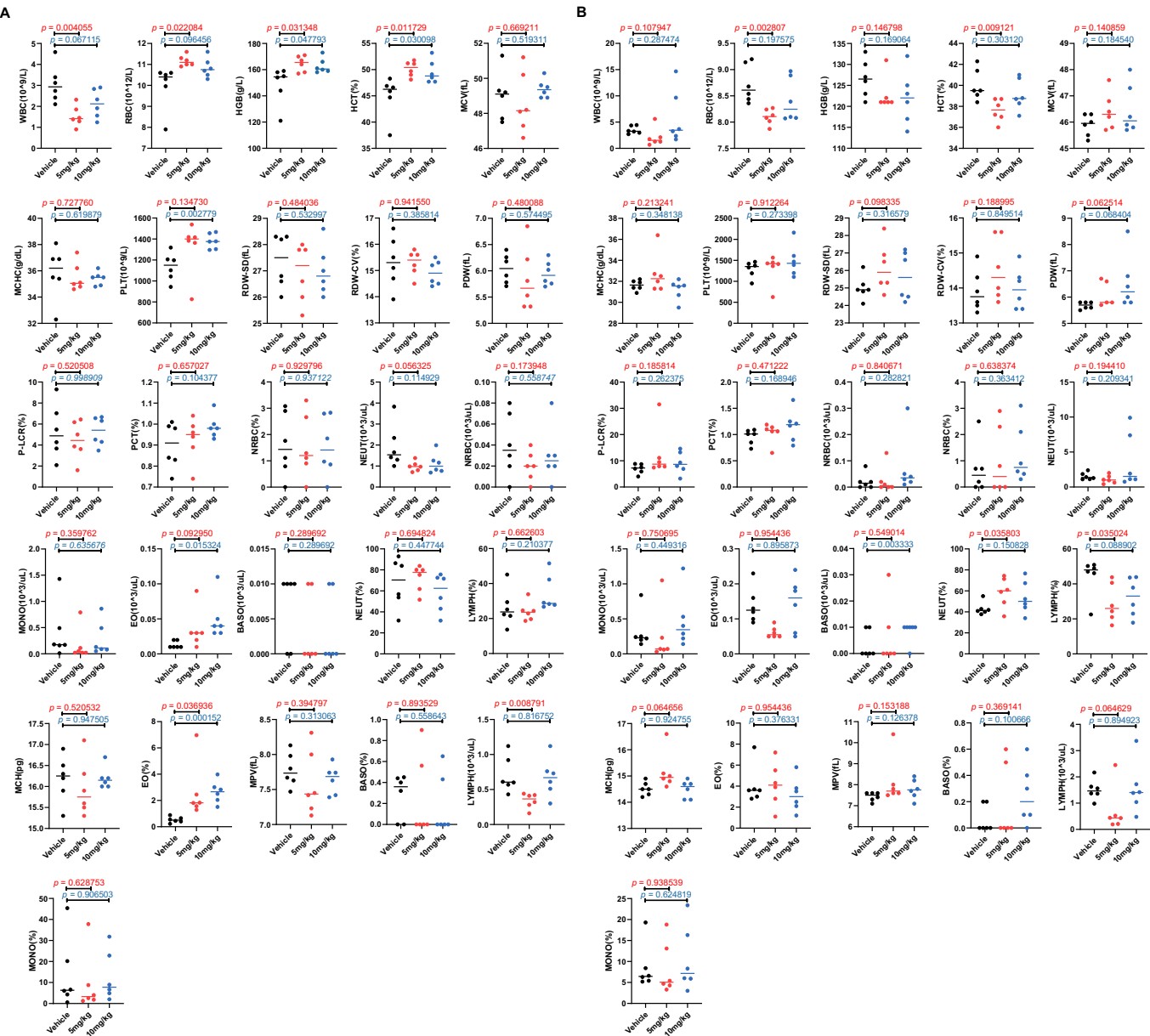

**Figure EV3.  RP-6306, a PKMYT1 inhibitor, suppresses tumor growth and proliferation in PDAC CDX and PDX models.**

Routine peripheral blood tests show no significant differences in the RP-6306-treated mice bearing Pa-Tu-8988T xenografts (**A**) ($n = 6$ per group) and PDXs (**B**) ($n = 6$ per group). WBC white blood cell, RBC red blood cell, HGB hemoglobin, HCT hematocrit, MCV mean corpuscular volume, MCH mean corpuscular hemoglobin, MCHC mean corpuscular-hemoglobin concentration, PLT platelet, PDW platelet distribution width, MPV mean platelet volume, P-LCR platelet-large cell ratio, PCT plateletcrit, NRBC nucleated red blood cells, NEUT neutrophil, LYMPH lymphocyte, MONO monocyte, EO eosinophil, BASO basophile. Unpaired $t$ test.

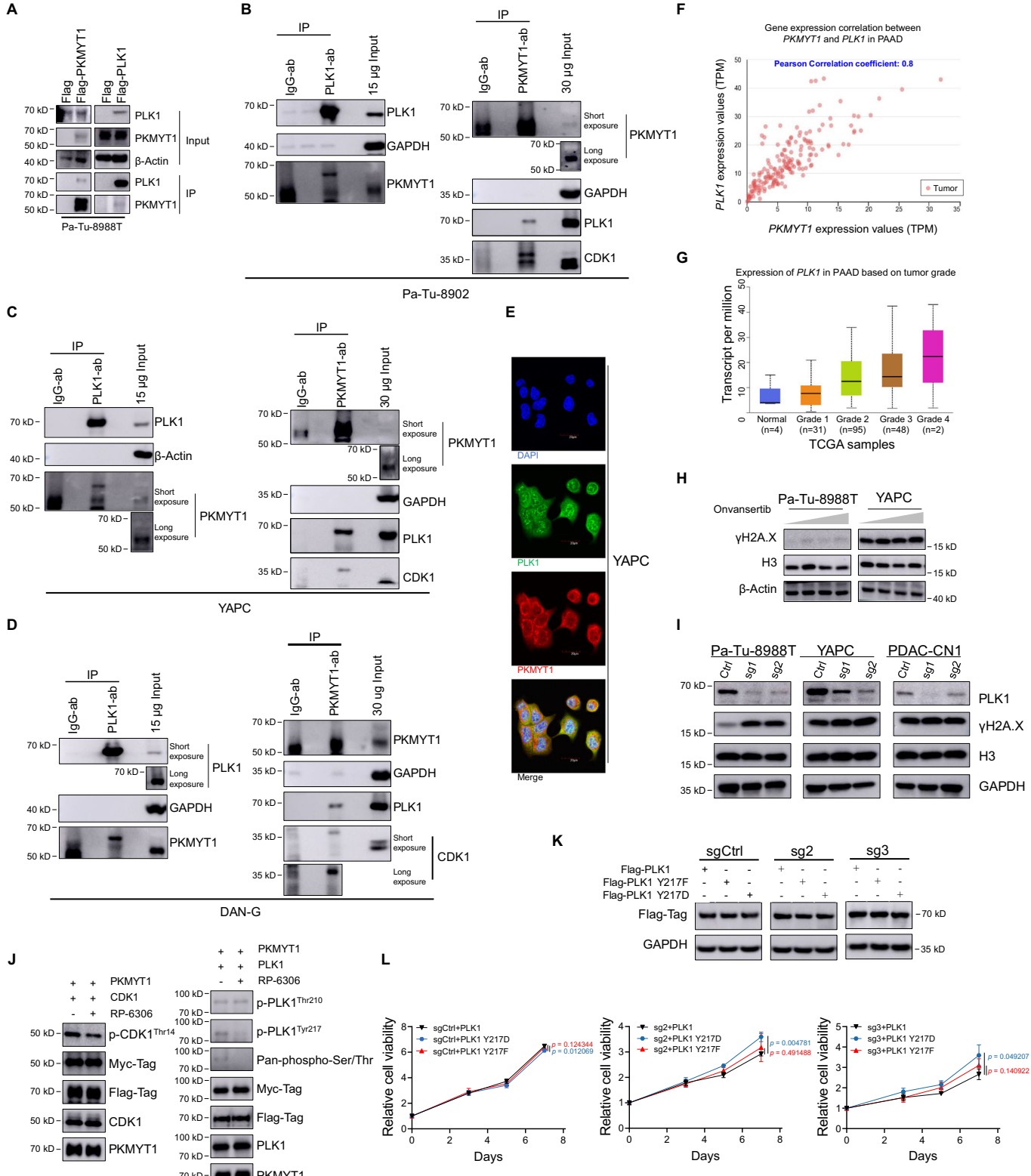

◄ **Figure EV4. PKMYT1 interacts with PLK1 and regulates PLK1 expression and phosphorylation.**

(A) PLK1 interacts with PKMYT1 in Pa-Tu-8988T. The cells were transduced with Flag-PKMYT1, Flag-PLK1 or Flag lentivirus, and cell lysates were subjected to a co-IP assay using an anti-Flag antibody, followed by immunoblotting with the indicated antibodies. (B–D) The endogenous PLK1-PKMYT1 interaction is confirmed in YAPC, Pa-Tu-8902 and DAN-G cells by a co-IP assay. (E) Immunofluorescence assay shows that PKMYT1 protein distributes in the cytoplasm, and PLK1 protein spreads in nucleus and cytoplasm. Cytoplasm PKMYT1 and PLK1 overlap. Scale bars: 20 μm. (F) Positive correlation between the expression levels of *PKMYT1* and *PLK1* ($n = 179$). (G) *PLK1* expression in human pancreatic cancer tissues with different grades (Normal, $n = 4$; Grade 1, $n = 31$; Grade 2, $n = 95$; Grade 3, $n = 48$; Grade 4, $n = 2$). The low bound, centerline, and upper bound of boxplot represent the first quartile, the median, and the third quartile of data, respectively; the upper and lower whiskers extend to the largest and smallest value. (H) Onvansertib (PLK1 inhibitor) treatment has no effect on DNA damage accumulation. Concentrations of compound used are listed in "Methods". (I) Knockout of *PLK1* (sg1 and sg2) does not induce DNA damage accumulation in all cell lines. (J) Phosphorylation signals were revealed using western blotting (Pan-phospho-Ser/Thr antibody, Phospho-PLK1-Thr210 antibody; Phospho-PLK1–Tyr217 antibody). Myc-CDK1 phosphorylation by PKMYT1 was used as a control. Western blotting analysis indicates PKMYT1 phosphorylates PLK1 Tyr217 and other Ser/Thr sites except Thr210. (K, L) CellTiter-Glo viability assay of the indicated YAPC cells transduced with lentiviral vectors expressing sgRNA targeting *PKMYT1* (sg2 or sg3) or control sgRNA (sgCtrl) along with phospho-mimicking PLK1 Tyr217Asp (Y217D) mutant, Tyr217Phe mutant (Y217F, which can not be phosphorylated) or wild-type PLK1. Tyr217 phosphorylation is involved in the oncogenic function of PKMYT1. The error bars indicate the mean ± s.d. of three replicates; unpaired *t* test.

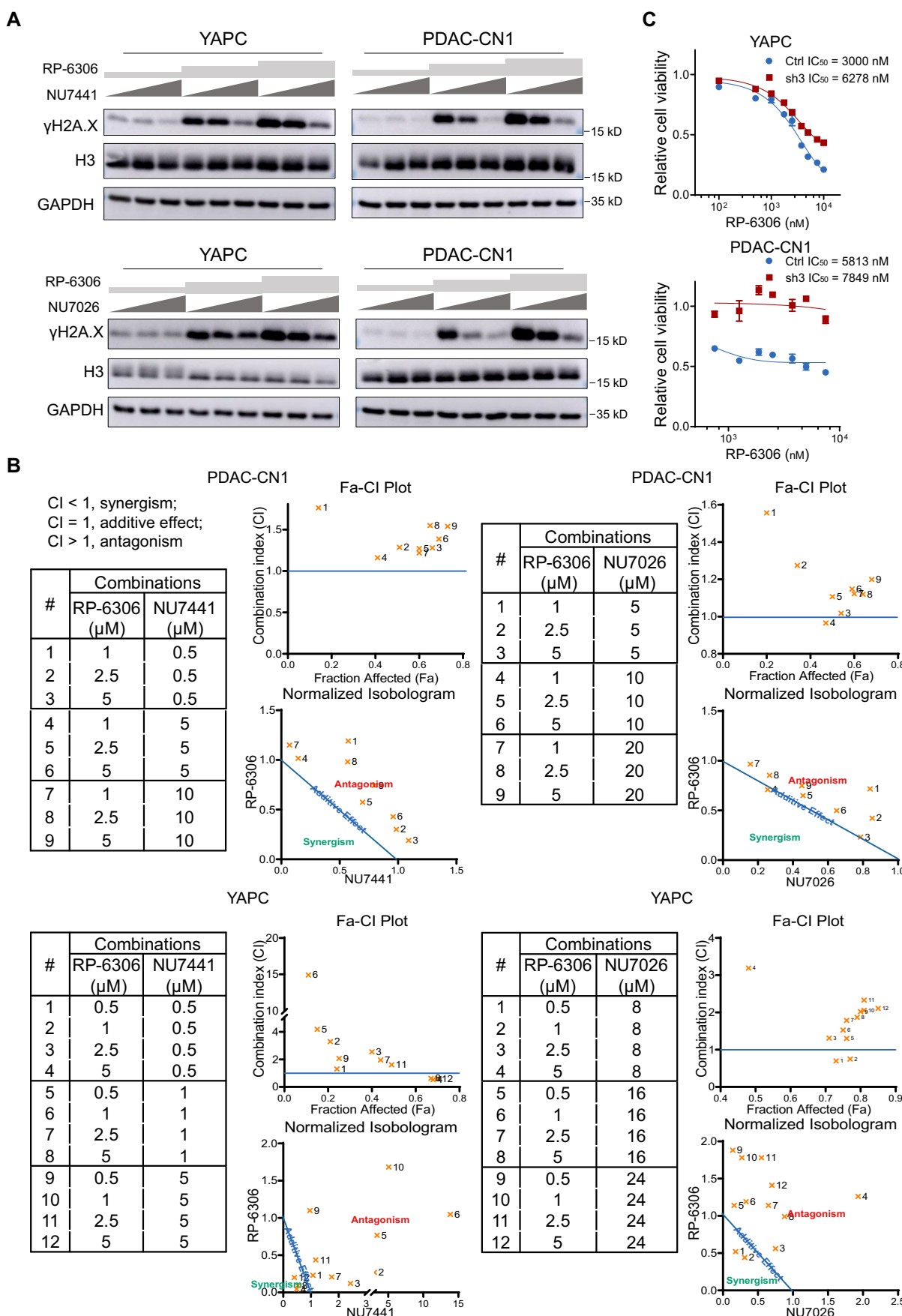

**Figure EV5.  PKMYT1 ablation activates PRKDC, whose activity modulates the sensitivity of PDAC to PKMYT1 inhibition.**

(A) The PRKDC inhibitor NU7441 and NU7026 disturb γH2A.X accumulation induced by the RP-6306 treatment. As the concentration of RP-6306 (lane 1, 4, and 7) increases, γH2AX accumulation increases. For each dosage of RP-6306 (lane 4–6, and lane 7–9), γH2AX accumulation decreases in a NU7441/NU7026 concentration-dependent manner, showing that PRKDC is involved in DNA damage response induced by PKMYT1 inhibition. Concentrations of inhibitors are listed in "Methods". (B) The PRKDC inhibitor NU7441 (left) and NU7026 (right) modulate the sensitivity of PDAC to PKMYT1 inhibitor. CI (top) and isobologram (bottom) analyses reveal the antagonistic effect between PKMYT1 inhibitor and PRKDC inhibitors in PDAC (CI > 1). Representative Fa-CI plots (top) and normalized isobolograms (bottom) are shown ($n = 3$ per group). (C) *PRKDC* knockdown (sh3) desensitizes cells to RP-6306. $IC_{50}$ values are shown. The error bars indicate the mean ± s.e.m. of three replicates.

