## [Peer Review File · EMBO Molecular Medicine]

Genome-wide CRISPR screens identify PKMYT1 as a therapeutic target in pancreatic ductal adenocarcinoma

Simin Wang, Yangjie Xiong, Yuxiang Luo, Yanying Shen, Fengrui Zhang, Haoqi Lan, Yuzhi Pang, Xiaofang Wang, Xiaoqi Li, Xufen Zheng, Xiaojing Lu, Xiaoxiao Liu, Yumei Cheng, Tanwen Wu, Yue Dong, Yuan Lu, Jiujiu Cui, Xiaona Jia, Sheng Yang, Liwei Wang, and Yuexiang Wang

Corresponding authors: Yuexiang Wang (yxwang76@sibs.ac.cn) , Liwei Wang (liweiwang@shsmu.edu.cn)

Review Timeline:

Submission Date:	25th Oct 23
Editorial Decision:	14th Nov 23
Revision Received:	12th Feb 24
Editorial Decision:	1st Mar 24
Revision Received:	10th Mar 24
Accepted:	14th Mar 24

Editor: Lise Roth

Transaction Report:

14th Nov 2023

Dear Dr. Wang,

Thank you for submitting your work to EMBO Molecular Medicine. We have now heard back from the referees who agreed to evaluate your manuscript. As you will see below, the reviewers find that the question addressed by the study is of potential interest, however they remain unconvinced that some of the major conclusions are sufficiently supported by the data.

As addressing all referees' concerns would require a lot of additional work and experiments, we further consulted with the referees, who agreed that the focus of the revisions should be on experiments advancing the understanding of the interplay between PKMYT1 and PLK1. They also agreed that the role of this interaction in DNA damage response could be addressed in writing only. Please note that all minor concerns must be addressed. If you feel you can satisfactorily address these points, you may wish to submit a revised version of your manuscript. Please attach a covering letter giving details of the way in which you have handled each of the points raised by the referees. A revised manuscript will once again be subject to review, and we cannot guarantee at this stage that the eventual outcome will be favorable.

We are expecting your revised manuscript within three months, if you anticipate any delay, please contact us.

We require:

- 1) A .docx formatted version of the manuscript text (including legends for main figures, EV figures and tables). Please make sure that the changes are highlighted to be clearly visible.
- 2) Individual production quality figure files as .eps, .tif, .jpg (one file per figure). For guidance, download the 'Figure Guide PDF' (<https://www.embopress.org/page/journal/17574684/authorguide#figureformat>).
- 3) At EMBO Press we ask authors to provide source data for the main figures. Our source data coordinator will contact you to discuss which figure panels we would need source data for and will also provide you with helpful tips on how to upload and organize the files.
- 4) A .docx formatted letter INCLUDING the reviewers' reports and your detailed point-by-point responses to their comments. As part of the EMBO Press transparent editorial process, the point-by-point response is part of the Review Process File (RPF), which will be published alongside your paper.
- 5) A complete author checklist, which you can download from our author guidelines (<https://www.embopress.org/page/journal/17574684/authorguide#submissionofrevisions>). Please insert information in the checklist that is also reflected in the manuscript. The completed author checklist will also be part of the RPF.
- 6) Please note that all corresponding authors are required to supply an ORCID ID for their name upon submission of a revised manuscript. An ORCID identified is currently missing for Liwei Wang.
- 7) It is mandatory to include a 'Data Availability' section after the Materials and Methods. Before submitting your revision, primary datasets produced in this study need to be deposited in an appropriate public database, and the accession numbers and database listed under 'Data Availability'. Please remember to provide a reviewer password if the datasets are not yet public (see <https://www.embopress.org/page/journal/17574684/authorguide#dataavailability>). Note that the Data Availability Section is restricted to new primary data that are part of this study.
- 8) For data quantification: please specify the name of the statistical test used to generate error bars and P values, the number (n) of independent experiments (specify technical or biological replicates) underlying each data point and the test used to calculate p-values in each figure legend. The figure legends should contain a basic description of n, P and the test applied. Graphs must include a description of the bars and the error bars (s.d., s.e.m.). Please provide exact p values.
- 9) Our journal encourages inclusion of *data citations in the reference list* to directly cite datasets that were re-used and obtained from public databases. Data citations in the article text are distinct from normal bibliographical citations and should directly link to the database records from which the data can be accessed. In the main text, data citations are formatted as

follows: "Data ref: Smith et al, 2001" or "Data ref: NCBI Sequence Read Archive PRJNA342805, 2017". In the Reference list, data citations must be labeled with "[DATASET]". A data reference must provide the database name, accession number/identifiers and a resolvable link to the landing page from which the data can be accessed at the end of the reference. Further instructions are available at .

13) Author contributions: CRediT has replaced the traditional author contributions section because it offers a systematic machine readable author contributions format that allows for more effective research assessment. Please remove the Authors Contributions from the manuscript and use the free text boxes beneath each contributing author's name in our system to add specific details on the author's contribution. More information is available in our guide to authors.

16) As part of the EMBO Publications transparent editorial process initiative (see our Editorial at <http://embomolmed.embopress.org/content/2/9/329>), EMBO Molecular Medicine will publish online a Review Process File (RPF) to accompany accepted manuscripts.

In the event of acceptance, this file will be published in conjunction with your paper and will include the anonymous referee reports, your point-by-point response and all pertinent correspondence relating to the manuscript. Let us know whether you agree with the publication of the RPF and as here, if you want to remove or not any figures from it prior to publication. Please note that the Authors checklist will be published at the end of the RPF.

I look forward to receiving your revised manuscript.

Yours sincerely,

Lise Roth

***** Reviewer's comments *****

Referee #1 (Remarks for Author):

The study provides evidence for a role for PKMYT1 in the regulation of pancreatic cancer tumor growth. Despite the fact that oncogenic role for this kinase in different tumor types, some of the findings carry disease significance for PDAC. The experiments are for the most part well conducted and clearly presented, however, additional experiments are needed to support the main conclusions of the study (specifically, the role of PPLK1 in the regulation of PKMYT1's protumoral function). First, the endogenous IP of Supp Fig S7B should be improved, the blots presented does not confidently showed the interaction. The IP should be done in other lines, validated using a second antibody for PKMYT1, and PLA assay. Also, the reversed IP using PLK1 antibodies should done as well. In addition, the endogenous IP should include positive controls with known interactors of PKMYT1 and PLK1. Second, the authors should define if PKMYT1 phosphorylates PLK1 (or vice versa). If the either of the events is true, the authors should determine if the phosphorylation sites are essential for the oncogenic function of these kinases using mutant that can't be phosphorylated. Preferable these experiments should be done using in vivo models. If changes in phosphorylation are not involved in the oncogenic function of PKMYT1, the authors should define if the interaction affects the enzymatic activity of these kinases by looking phosphorylation of known substrates. Third, the effect of DNA damage is not well defined and aligned with the main findings of the study. How the changes in DNA damage play a role in the oncogenic function of PKMYT1 (and the PKMYT1-PLK1 interplay). This should be defined and rationale should be supported experimentally. Fourth, there are few minor (but important points should be addressed including a) the quality of PKMYT1 blots of Figure 4B, b) quantification of IHC of Figure 2A should be provided (also number of cases, and higher magnification pictures of multiple cases), and c) in all blots molecular markers flanking the band of interest should be added.

Referee #2 (Comments on Novelty/Model System for Author):

Interesting work, some caveats.

Referee #2 (Remarks for Author):

Interesting piece of work on which I have only few comments:

Figure 2E raises the question what mechanism governs high PKMYT1 expression? The 11% of cases shown here cannot explain it in full. For instance, the KMs in the same figure are split by median, meaning that half of the samples are considered PKMYT1-high.

Given that the constitutive loss of PKMYT1 hampers cell growth in vitro, what other outcome could possibly have been expected from the in vivo experiments? The cells were crippled at the moment of injection.

Which currently given cytotoxic does the RP-6306 drug synergise with most?

The observations on TP53 status reported at the end are a bit confusing in light of the prevalence of TP53 mutations in patients and association of PKMYT1 with outcome. What do the analyses of clinical outcome parameters look like when TP53 status is taken into account.

MINOR COMMENTS

The authors mention that a minority of PDAC cases have a targetable mutation but with the advent of G12D/V inhibitors, this statement is obsolete.

"Inhibitor development and functional characterization of PKMYT1 has severely lagged" is bit of an odd statement given that the

authors needed a CRISPR screen to identify it.

Figure 1A; I'd say these correlations are rather weak.

IHC in Figure 2A does not look very specific. There is a lot of background staining.

Figure 2; was expression correlated with proliferative index?

Referee #3 (Comments on Novelty/Model System for Author):

Both the human PDAC cell models and in vivo PDX models used in this study are commonly used by most researchers in the study of pancreatic cancer.

Referee #3 (Remarks for Author):

The study "Genome-wide CRISPR screens identify PKMYT1 as a therapeutic target in pancreatic ductal adenocarcinoma" by Wang and colleagues employs a genome-wide CRISPR loss-of-function screen and identifies PKMYT1 as an oncogenic driver and potential prognostic biomarker in pancreatic ductal adenocarcinoma (PDAC). The ultimate goal of this study is the characterization of genes that might constitute suitable targets for a rationale therapy in PDAC. Through multiple in vitro and in vivo functional studies, the authors further validate the PKMYT1 as a potential target for pharmacological intervention using the available compounds. Finally, they show that PKMYT1 promotes PDAC tumorigenesis by regulating PLK1 expression. TP53 mutation status and PRKDC activation modulate the sensitivity to PKMYT1 inhibition. These results well define PKMYT1 dependency in PDAC and identify potential therapeutic strategies for clinical translation. Overall, the study is interesting, nicely written and well conducted, will have a significant impact on both PDAC research and treatment. A few comments may strength the manuscript.

1. More than 80% of PDAC harbor oncogenic KRAS mutation, serving as an attractive therapeutic target in the PDAC (Strickler, Satake et al., 2023). Two representative human pancreatic adenocarcinoma cell lines were used in the CRISPR screens. Do the two cell lines contain KRAS mutation? If so, does the two cell models show KRAS dependency? In addition, rank-ordered med. CS for SDHC and COX7B (two hits highlighted in the Supplementary Fig. S2) from the screens should be indicated in the Figure 1E.
2. Figure 2E does not seem like it needs to be displayed as a figure but can rather be stated in the text.
3. For the crystal violet assays (Figure 3C) and the anchorage-independent growth assays (Figure 3D), some of the control cells (such as YAPC) have such low seeding density that it is hard to make a solid conclusion. This may be helpful to repeat with a higher starting seeding density.
4. Drugs IC50 in Figure 4A, Supplementary Fig. S5B and Supplementary Fig. S6B should be reported in this study.
5. Could the authors quantify the MG132 western blots results in Figure 6F, just as the results showing in the Figure 6E?
6. For the Figure 7 and Supplementary Fig. S8, the combination experiment in the cell line panel was carried out using a range of concentrations of RP-6306 and PRKDC inhibitors. Could the authors provide the drugs IC50 of PRKDC inhibitors in the PDAC cells (Pa-Tu-8988T, YAPC and PDAC-CN1) in the Figure 7 and Supplementary Fig. S8? This would be more informative.
7. Panel Supplementary Fig. S8 and Supplementary Fig. S9 should be cited in the text before Supplementary Fig. S10.

Referee #1 (Remarks for Author):

The study provides evidence for a role for PKMYT1 in the regulation of pancreatic cancer tumor growth. Despite the fact that oncogenic role for this kinase in different tumor types, some of the findings carry disease significance for PDAC. The experiments are for the most part well conducted and clearly presented, however, additional experiments are needed to support the main conclusions of the study (specifically, the role of PPLK1 in the regulation of PKMYT1's protumoral function).

We thank the reviewer for the thoughtful comments. We are glad that the reviewer found that “some of the findings carry disease significance for PDAC. The experiments are for the most part well conducted and clearly presented”. We have addressed these constructive comments in the manuscript as indicated below:

First, the endogenous IP of Supp Fig S7B should be improved, the blots presented does not confidently showed the interaction. The IP should be done in other lines, validated using a second antibody for PKMYT1, and PLA assay. Also, the reversed IP using PLK1 antibodies should done as well. In addition, the endogenous IP should include positive controls with known interactors of PKMYT1 and PLK1.

Thank you for the helpful comments! With the new experiments, the endogenous PLK1-PKMYT1 interaction has been confirmed in THREE PDAC cell lines (YAPC, Pa-Tu-8902 and DAN-G) (NEW Fig EV4B-D). The antibodies used for IP is different from those used for western blotting validation. For example, we used the first PKMYT1 antibody (Santa Cruz #sc-74523) for endogenous IP. The second PKMYT1 antibody (Cell Signaling Technology #4282S) was used for western blotting validation. Also, the reversed IP used two PLK1 antibodies (ABMART #MG670393 and Abclonal #A21082) as well. CDK1 works as a positive control (NEW Fig EV4B-D). It's regret that a commercial kit for PLA assay (Merck #DUO92102) is not available because of the limited timeframe and the challenge to buy the reagents from the oversea companies, as *EMBO Mol Med* has requested us to submit the revised manuscript by 13-Feb-2024. But we share the comments that the interaction between PKMYT1 and PLK1 should be validated rigorously. The *in vitro* kinase assay presented bellow (for the second comment, NEW Fig EV4J) can confidently show the direct interaction. In addition, another new experiment - immunofluorescence assay - was performed. We show that although PLK1 protein spreads in nucleus, both PKMYT1 and PLK1 protein distributed in the cytoplasm where they may interact with each other (NEW Fig EV4E) indicated by the overlapped signals. All the NEW experimental evidences clearly demonstrate the interaction between PKMYT1 and PLK1.

These data are reported in the manuscript: NEW Fig EV4B-E; NEW Fig EV4J.

Second, the authors should define if PKMYT1 phosphorylates PLK1 (or vice versa). If the either of the events is true, the authors should determine if the phosphorylation sites are essential for the oncogenic function of these kinases using mutant that can't be phosphorylated. Preferable these experiments should be done using in vivo models. If changes in phosphorylation are not involved in the oncogenic function of PKMYT1, the authors should define if the interaction affects the enzymatic activity of these kinases by looking phosphorylation of known substrates.

We thank the reviewer for the constructive comment. To investigate whether PKMYT1 phosphorylates PLK1, an *in vitro* incubation of PLK1 immunoprecipitates with purified recombinant PKMYT1 enzyme in the presence of ATP was performed to confirm a direct phosphorylation of PLK1 by PKMYT1 (NEW Fig EV4J; NEW Appendix Supplementary Materials and Methods - *In vitro* kinase assay) as reported previously (Gelot et al., 2023, Moon et al., 2017).

As *PKMYT1* gene encodes a member of the serine/threonine protein kinase family, we observed PLK1 phosphorylation at pan-Ser/Thr sites when incubated with PKMYT1. The phosphorylation was markedly reduced when reaction system was added with PKMYT1 inhibitor (NEW Fig EV4J). Similarly, as PKMYT1 regulates phosphorylation of CDK1-Thr14 and Tyr15, we focused on two reported residues Thr210 and Tyr217, which are functionally important (Caron, Byrne et al., 2016, Macurek, Lindqvist et al., 2008). We identified a PKMYT1 phosphorylation site on PLK1 that the phosphorylation of PLK1-Tyr217 was markedly increased when incubated with PKMYT1 compared with PKMYT1 kinase inhibited by RP-6306 (NEW Fig EV4J). Next, to determine if the phosphorylation site is essential for the oncogenic function of PKMYT1, different mutants were constructed. Consistently, phospho-mimicking PLK1 Tyr217Asp mutant but not Tyr217Phe mutant (which can't be phosphorylated) overexpression attenuated the growth and proliferation inhibition properties in PKMYT1-ablated PDAC cells (NEW Fig EV4K-L). Together, our results show that PKMYT1 phosphorylates PLK1 at Tyr217 and other potential Ser/Thr sites. Tyr217 phosphorylation is involved in the oncogenic function of PKMYT1 although we can not rule out whether other Ser/Thr residues are also involved. The detailed mechanisms between PKMYT1 and PLK1 merit further investigation.

These data are reported in the manuscript: NEW Fig EV4J-L; NEW Appendix Supplementary Materials and Methods - *In vitro* kinase assay.

Response references:

Caron D, Byrne DP, Thebault P, Soulet D, Landry CR, Evers PA, Elowe S (2016) Mitotic phosphotyrosine network analysis reveals that tyrosine phosphorylation regulates Polo-like kinase 1 (PLK1). *Sci Signal* 9: rs14

Gelot C, Kovacs MT, Miron S, Mylne E, Haan A, Boeffard-Dosierre L, Ghoul R, Popova T, Dingli F, Loew D, Guirouilh-Barbat J, Del Nery E, Zinn-Justin S, Ceccaldi R (2023) Poltheta is phosphorylated by PLK1 to repair double-strand breaks in mitosis. *Nature* 621: 415-422

Macurek L, Lindqvist A, Lim D, Lampson MA, Klompaker R, Freire R, Clouin C, Taylor SS, Yaffe MB, Medema RH (2008) Polo-like kinase-1 is activated by aurora A to promote checkpoint recovery. *Nature* 455: 119-23

Moon S, Kim J, Jho EH (2017) In vitro NLK Kinase Assay. *Bio Protoc* 7: e2593

Third, the effect of DNA damage is not well defined and aligned with the main findings of the study. How the changes in DNA damage play a role in the oncogenic function of PKMYT1 (and the PKMYT1-PLK1 interplay). This should be defined and rationale should be supported experimentally.

We thank the reviewer for the thoughtful comments, which allowed us to elaborate further in the discussion (also per Editor's comment: They also agreed that the role of this interaction in DNA damage response could be addressed in writing only). Studies shown that loss of either kinase interferes with the G2/M checkpoint, driving cells into mitosis prematurely (Chow & Poon, 2013). Loss of PKMYT1 dramatically influences the mitotic index of glioblastoma and human neural progenitors (Toledo, Ding et al., 2015), and a similar phenotype is observed in HeLa cells knockdown of PKMYT1 by siRNA (Villeneuve, Scarpa et al., 2013). Many of the CCNE1-high cells could skip G2 in response to PKMYT1 inhibition and do not go through a normal cell division but rather toggled between mitotic and interphase before terminating with high pan- γ H2AX signal (Gallo, Young et al., 2022). PKMYT1 knockout or RP-6306 treatment inhibits cell cycle progression (Fig 3H; Fig 4D; Fig EV2F) and induces pan- γ H2AX in PDAC cell lines (Fig 3I; revised Fig 4C; revised Fig EV2C), which indicated that G2/M checkpoint dysfunction caused by PKMYT1 inhibition leads to unchecked premature mitotic entry, then results in the accumulation of genetic lesions from unrepaired DNA damage, ultimately leading to apoptosis or mitotic catastrophe (Fig 3J; Fig 4E; Fig EV2G) (Asquith, Laitinen et al., 2020, Gallo et al., 2022).

The above summary has been incorporated in the discussion.

Response references:

Asquith CRM, Laitinen T, East MP (2020) PKMYT1: a forgotten member of the WEE1 family. *Nat Rev Drug Discov* 19: 157

Chow JP, Poon RY (2013) The CDK1 inhibitory kinase MYT1 in DNA damage checkpoint recovery. *Oncogene* 32: 4778-88

Gallo D, Young JTF, Fourtounis J, Martino G, Alvarez-Quiñon A, Bernier C, Duffy NM, Papp R, Roulston A, Stocco R, Szychowski J, Veloso A, Alam H, Baruah PS, Fortin AB, Bowlan J, Chaudhary N, Desjardins J, Dietrich E, Fournier S et al. (2022) CCNE1 amplification is synthetic lethal with PKMYT1 kinase inhibition. *Nature* 604: 749-756
Toledo CM, Ding Y, Hoellerbauer P, Davis RJ, Basom R, Girard EJ, Lee E, Corrin P, Hart T, Bolouri H, Davison J, Zhang Q, Hardcastle J, Aronow BJ, Plaisier CL, Baliga NS, Moffat J, Lin Q, Li XN, Nam DH et al. (2015) Genome-wide CRISPR-Cas9 Screens Reveal Loss of Redundancy between PKMYT1 and WEE1 in Glioblastoma Stem-like Cells. *Cell Rep* 13: 2425-2439
Villeneuve J, Scarpa M, Ortega-Bellido M, Malhotra V (2013) MEK1 inactivates Myt1 to regulate Golgi membrane fragmentation and mitotic entry in mammalian cells. *Embo J* 32: 72-85

Fourth, there are few minor (but important points should be addressed including a) the quality of PKMYT1 blots of Figure 4B,

RP-6306 treatment caused hyperphosphorylation of PKMYT1 indicated by drastically decreased mobility (Chow & Poon, 2013). The mobility was made qualitative and relative quantitative assessments in the Figure 4B (revised Fig 4B).

b) quantification of IHC of Figure 2A should be provided (also number of cases, and higher magnification pictures of multiple cases)

Thank you for the helpful comments! We updated the related parts. The quantification of IHC has been provided (revised Fig 2A-B)

c) in all blots molecular markers flanking the band of interest should be added.

We thank the reviewer for the careful review. The molecular markers have been added (revised Fig 3A; revised Fig 4B-C; revised Fig 6A; revised Fig 6C-F; revised Fig 7A; revised Fig EV2A; revised Fig EV2C-D; revised Fig EV4A-D; revised Fig EV4H-K; revised Fig EV5A; revised Appendix Fig S2A; revised Appendix Fig S2C; revised Appendix Fig S3A). Thank you!

Referee #2 (Comments on Novelty/Model System for Author):

Interesting work, some caveats.

Referee #2 (Remarks for Author):

Interesting piece of work on which I have only few comments:

We are glad that the reviewer found that our work is interesting. We thank the reviewer for the helpful and constructive comments.

Figure 2E raises the question what mechanism governs high PKMYT1 expression? The 11% of cases shown here cannot explain it in full. For instance, the KMs in the same figure are split by median, meaning that half of the samples are considered PKMYT1-high.

We thank the reviewer for the thoughtful comments, which allowed us to elaborate further in the discussion. PDAC displays elevated levels of PKMYT1 expression as the expression is higher in human pancreatic cancer tissues compared with noncancerous tissues from the TCGA-PAAD cohort (Fig 2C). The expression level of PKMYT1 progressively increased with increasing degree of tumor differentiation (Fig 2D). Positive expression was also validated in a human tissue microarray consisting of 75 patients with PDAC (36.0%) (revised Fig 2A-B). *PKMYT1* amplification was identified in 11% (12 of 109) of patients with PDAC in the UTSW cohort, consistent with the idea that genomic amplification of the *PKMYT1* locus in PDAC accounts for the high expression of PKMYT1. While gene amplification accounts for some of these high expression levels, it is not a widespread phenomenon. Studies have revealed that the downregulation of the demethylase *ALKBH5* leads to increased expression of *PKMYT1* in gastric cancer (Hu, Gong et al., 2022). Further exploration into the epigenetic modifications of the *PKMYT1* gene could provide insights into the mechanisms underlying its elevated expression, going beyond genomic amplifications.

Analyses with three independent cohorts also confirmed that higher PKMYT1 expression correlates with a worse prognosis in PDAC patients (Fig 2E). Kaplan–Meier curves for overall survival show that PDAC patients with high PKMYT1 expression (split by median) have worse prognoses than those with low PKMYT1 expression, which do not mean that half of the samples are PKMYT1-high. Actually, other grouping method such as divide the cohorts into three groups also shows that patients with high PKMYT1 expression have worse prognoses (Response Figure 1).

Response references:

Hu Y, Gong C, Li Z, Liu J, Chen Y, Huang Y, Luo Q, Wang S, Hou Y, Yang S, Xiao Y (2022) Demethylase ALKBH5 suppresses invasion of gastric cancer via PKMYT1 m6A modification. *Mol Cancer* 21: 34

Response Figure 1. Kaplan–Meier curves for overall survival show that PDAC patients with high PKMYT1 expression have worse prognoses than those with low PKMYT1 expression in GEO datasets GSE71729, GSE21501 and the TCGA pancreatic adenocarcinoma dataset. (Note that the sample size in TCGA with clinical parameters is now 177, not 170 in the former submission).

Given that the constitutive loss of PKMYT1 hampers cell growth in vitro, what other outcome could possibly have been expected from the in vivo experiments? The cells were crippled at the moment of injection.

Thank you for the helpful comments! We entirely agree that in vivo KO experiments couldn't rule out the possibility that the cells were crippled at the moment of injection and do not distinguish effects on initiation with tumor growth. To address this question, we have validated PKMYT1 function by using specific inhibitor, ie, the cells were not crippled at the moment of injection. RP-6306 treatment suppresses tumor growth and proliferation in PDAC cell line xenografts and PDX models *in vivo* (Fig 5). Thank you!

Which currently given cytotoxic does the RP-6306 drug synergise with most?

We thank the reviewer for the constructive comment. New experiments have been performed. PKMYT1 depletion sensitizes pancreatic cancer cells to gemcitabine (NEW Fig EV2J). PRKDC activation promotes PKMYT1 inhibition-induced cytotoxicity in PDAC (Fig 7; Fig EV5).

These data are reported in the manuscript: Fig 7; Fig EV2J; Fig EV5.

The observations on TP53 status reported at the end are a bit confusing in light of the prevalence of TP53 mutations in patients and association of PKMYT1 with outcome. What do the analyses of clinical outcome parameters look like when TP53 status is taken into account.

We thank the reviewer for the careful review. CRISPR genome editing tool was used to completely destroy the function of *TP53* in our manuscript. While *TP53* mutational spectrum, dominated by missense mutations in patients, are discrepant in diverse contexts with different studies such as loss-of-function, gain-of-function, or dominant-

negative effect (Boettcher, Miller et al., 2019, Chen, Zhang et al., 2022). It's not rational to analyze the relationship between *PKMYT1* and *TP53* based on mutation status. So the former Figure 8B was removed in the revised manuscript. Neither all kinds of *TP53* mutational spectrum are taken into account nor *TP53*-null frame shift mutations distinguish poor prognosis (Response Figure 2).

Response references:

Boettcher S, Miller PG, Sharma R, McConkey M, Leventhal M, Krivtsov AV, Giacomelli AO, Wong W, Kim J, Chao S, Kurppa KJ, Yang X, Milenkovic K, Piccioni F, Root DE, Rucker FG, Flamand Y, Neuberg D, Lindsley RC, Janne PA et al. (2019) A dominant-negative effect drives selection of *TP53* missense mutations in myeloid malignancies. *Science* 365: 599-604

Chen X, Zhang T, Su W, Dou Z, Zhao D, Jin X, Lei H, Wang J, Xie X, Cheng B, Li Q, Zhang H, Di C (2022) Mutant p53 in cancer: from molecular mechanism to therapeutic modulation. *Cell Death Dis* 13: 974

Response Figure 2. Kaplan–Meier curves for overall survival in the indicated TCGA pancreatic adenocarcinoma dataset.

MINOR COMMENTS

The authors mention that a minority of PDAC cases have a targetable mutation but with the advent of G12D/V inhibitors, this statement is obsolete.

We thank the reviewer for pointing this out, and we have revised the sentences in the Abstract and Introduction.

"Inhibitor development and functional characterization of PKMYT1 has severely lagged" is bit of an odd statement given that the authors needed a CRISPR screen to identify it.

We thank the reviewer for pointing this out. The statement has been removed.

Figure 1A; I'd say these correlations are rather weak.

The entire GeCKOv2 library can be delivered as two half-libraries (A and B). Each one has approximately 3 sgRNAs targeting different genome sequence. We entirely agree with the reviewer's comment that the correlations are not strong, although statistically significant. Therefore, the potential candidates were identified carefully and validated rigorously. In addition, two PDAC cell lines were screened (Fig 1B) and two different algorithms were adopted (Fig EV1C) to improve the accuracy of the screens. We have revised the result as follows:

A correlation (the word "significant" has been removed) in the dependency scores (med.CS) was observed between the two libraries (Fig 1A)

IHC in Figure 2A does not look very specific. There is a lot of background staining.

We thank the reviewer for the careful review. We have repeated the IHC experiments (revised Fig 2A-B). IHC staining was performed with validated antibodies against PKMYT1. An interesting fact, PKMYT1 expresses in normal cells (although lower than that in tumor tissue), so the background does not look like completely negative.

Figure 2; was expression correlated with proliferative index?

PKMYT1 expression correlates with proliferative index indicated by Ki67 (NEW Fig EV2G).

Referee #3 (Comments on Novelty/Model System for Author):

Both the human PDAC cell models and in vivo PDX models used in this study are commonly used by most researchers in the study of pancreatic cancer.

Referee #3 (Remarks for Author):

The study "Genome-wide CRISPR screens identify PKMYT1 as a therapeutic target in pancreatic ductal adenocarcinoma" by Wang and colleagues employs a genome-wide CRISPR loss-of-function screen and identifies PKMYT1 as an oncogenic driver and potential prognostic biomarker in pancreatic ductal adenocarcinoma (PDAC). The ultimate goal of this study is the characterization of genes that might constitute suitable targets for a rationale therapy in PDAC. Through multiple in vitro and in vivo functional studies, the authors further validate the PKMYT1 as a potential target for pharmacological intervention using the available compounds. Finally, they show that PKMYT1 promotes PDAC tumorigenesis by regulating PLK1 expression. TP53 mutation status and PRKDC activation modulate the sensitivity to PKMYT1 inhibition. These results will define PKMYT1 dependency in PDAC and identify potential therapeutic strategies for clinical translation. Overall, the study is interesting, nicely written and well conducted, will have a significant impact on both PDAC research and treatment. A few comments may strengthen the manuscript.

We thank the reviewer for this favorable summary.

More than 80% of PDAC harbor oncogenic KRAS mutation, serving as an attractive therapeutic target in the PDAC (Strickler, Satake et al., 2023). Two representative human pancreatic adenocarcinoma cell lines were used in the CRISPR screens. Do the two cell lines contain KRAS mutation? If so, does the two cell models show KRAS dependency? In addition, rank-ordered med. CS for SDHC and COX7B (two hits highlighted in the Supplementary Fig. S2) from the screens should be indicated in the Figure 1E.

We thank the reviewer for the construction comments. Yes, both of the two cell lines contain oncogenic KRAS mutations (revised Table EV3) and show KRAS dependency (revised Fig 1E). In addition, rank-ordered med. CS for SDHC and COX7B has been indicated in the revised Fig 1E.

Figure 2E does not seem like it needs to be displayed as a figure but can rather be stated in the text.

Former Figure 2E has been removed and the information is presented in the Discussion as follow: PKMYT1 amplification was identified in 11% (12 of 109) of patients with PDAC in the UTSW cohort.

For the crystal violet assays (Figure 3C) and the anchorage-independent growth assays (Figure 3D), some of the control cells (such as YAPC) have such low seeding density that it is hard to make a solid conclusion. This may be helpful to repeat with a higher starting seeding density.

We thank the reviewer for the careful review. We have repeated the experiments and revised the Fig 3C and 3D.

Drugs IC50 in Figure 4A, Supplementary Fig. S5B and Supplementary Fig. S6B should be reported in this study.

A new Supplementary Table (NEW Table EV5) has been added to include the IC₅₀.

Could the authors quantify the MG132 western blots results in Figure 6F, just as the results showing in the Figure 6E?

Quantification of the MG132 western blots in the Figure 6F has been provided (revised Fig 6F).

For the Figure 7 and Supplementary Fig. S8, the combination experiment in the cell line panel was carried out using a range of concentrations of RP-6306 and PRKDC inhibitors. Could the authors provide the drugs IC50 of PRKDC inhibitors in the PDAC cells (Pa-Tu-8988T, YAPC and PDAC-CN1) in the Figure 7 and Supplementary Fig. S8? This would be more informative.

A new Supplementary Table (NEW Table EV5) has been added to include the IC₅₀.

Panel Supplementary Fig. S8 and Supplementary Fig. S9 should be cited in the text before Supplementary Fig. S10.

We have made the changes.

We would be delighted to respond to any additional criticisms that might arise in re-review of the manuscript.

We are most grateful for this opportunity to revise our manuscript.

1st Mar 2024

Dear Dr. Wang,

Thank you for submitting your revised manuscript. We have now received the feedback from the three referees who who re-reviewed your manuscript. As you will see below, they are satisfied with the revisions, and I will therefore be able to accept your manuscript once the following editorial points will be addressed:

1/ Please address the minor comments from referee #2.

2/Manuscript text:

- Your manuscript was cross-checked for text similarities, and a few matches were found with previously published material (please see screenshots attached). Please modify your text accordingly.
- Please remove the red text, and only keep in track changes mode any new modification.
- The manuscript sections should be in the following order: Title page - Abstract & Keywords - Introduction - Results - Discussion - Materials & Methods - Data Availability - Acknowledgments - Disclosure Statement & Competing Interests - References - Figure Legends - Tables with legends - Expanded View Figure Legends.
- The list of Expanded View Figures and Tables and Appendix (on p 12) needs to be removed from the manuscript.
- Materials and Methods:
 - o Please include the methods currently in Appendix Supplementary Materials and Methods in the main manuscript text.
 - o Human samples: please provide a sentence stating that the experiments conformed to the principles set out in the WMA Declaration of Helsinki and the Department of Health and Human Services Belmont Report.
 - o Antibodies: please provide dilutions/concentrations.
 - o Statistics: please provide a statement on sample size, inclusion/exclusion criteria, randomization, and blinding.
- Funding should be merged with Acknowledgements.
- Please replace "Competing Interests" by "Disclosure statement and competing interests". We updated our journal's competing interests policy in January 2022 and request authors to consider both actual and perceived competing interests. Please review the policy <https://www.embopress.org/competing-interests> and update your competing interests if necessary.
- References: please make sure to have 10 authors listed before et al.

3/ Figures and Appendix:

- Thank you for providing exact p values. Please also provide the exact values for n.s., non-significant.
- Tables EV1, EV2, EV4 and EV5 should be made Datasets EV1, etc. Table EV3 can remain an EV table (Table EV1).
- Appendix: please provide page numbers on the title page. The Materials and Methods should be moved to the main manuscript file. The information about EV figures and tables should be removed from the title page.
- Please make sure all figures/figure panels are referenced in the manuscript text: currently, a callout for Figure 4E is missing. The following callout needs to be corrected: Figure S3A.
- Please carefully check the composition of your figures, in particular Figure 6C and Figure EV2 C (GAPDH blots), as well as the pictures used in Figure 8E. Figure re-use must be mentioned in the figure legends.
- Please address the queries from our data editors in the figure legends:
 1. Please indicate the statistical test used for data analysis in the legends of figures 1a-c; 2e; 8a-b; EV 1c.
 2. Please note that the box plot needs to be defined in terms of minima, maxima, centre, bounds of box and whiskers, and percentile in the legends of figure 2c.
 3. Please note that the box plots need to be defined in terms of minima, maxima, bounds of box and whiskers in the legends of figure 2d; EV 4g.
 4. Please note that information related to n is missing in the legends of figure 2d; EV 1d.
 5. Although 'n' is provided, please describe the nature of entity for 'n' in the legends of figure 7d; EV 2b; EV 4g; EV 5c.

4/ Thank you for providing Source Data. Please upload them as one folder per figure.

5/ Checklist:

- Please complete the Statistics section.
- Please complete the section Ethics/ human participants (Helsinki declaration)
- Please correct the reference to where information is listed wherever needed.

6/ Please note that all corresponding authors are required to supply an ORCID ID for their name upon submission of a revised manuscript. An ORCID identifier is currently missing for Liwei Wang.

7/ Our journal encourages inclusion of *data citations in the reference list* to directly cite datasets that were re-used and obtained from public databases. Data citations in the article text are distinct from normal bibliographical citations and should directly link to the database records from which the data can be accessed. In the main text, data citations are formatted as follows: "Data ref: Smith et al, 2001" or "Data ref: NCBI Sequence Read Archive PRJNA342805, 2017". In the Reference list,

data citations must be labeled with "[DATASET]". A data reference must provide the database name, accession number/identifiers and a resolvable link to the landing page from which the data can be accessed at the end of the reference. Further instructions are available at .

8/ Synopsis:

Please reformat your synopsis text to the following format:

- a short stand first (maximum of 300 characters, including space)
- 2-5 one-sentences bullet points that summarize the paper (maximum of 30 words / bullet point).

Please resize your synopsis image 550 px wide x 300-600 px high and make sure that the text remains legible. Please provide a white background.

9/ As part of the EMBO Publications transparent editorial process initiative (see our Editorial at <http://embomolmed.embopress.org/content/2/9/329>), EMBO Molecular Medicine will publish online a Review Process File (RPF) to accompany accepted manuscripts.

This file will be published in conjunction with your paper and will include the anonymous referee reports, your point-by-point response and all pertinent correspondence relating to the manuscript. Let us know whether you agree with the publication of the RPF and as here, if you want to remove or not any figures from it prior to publication.

I look forward to receiving your revised manuscript.

Yours sincerely,

Lise Roth

Lise Roth, PhD

Senior Editor

EMBO Molecular Medicine

***** Reviewer's comments *****

Referee #1 (Comments on Novelty/Model System for Author):

The authors have been responsive to reviewers critiques and the main weaknesses have been adequately addressed.

Referee #1 (Remarks for Author):

No further comments.

Referee #2 (Remarks for Author):

The authors have addressed most of my concerns. Please find some minor remaining points:

>> Which currently given cytotoxic does the RP-6306 drug synergise with most?

> gemcitabine

I meant to ask whether the authors could show to which cytotoxic such synergy could be found. Have they also tested 5-FU, nab-paclitaxel, oxaliplatin, irinotecan?

Do note that something strange happened to Figure 1C.

Figure 2B could perhaps just be mentioned in the text?

Can the authors find a way to simplify the numerous significance values in Figure 4E? Perhaps a graphical way to plot these numbers?

It would help to harmonise font sizes across panels and figures.

Referee #3 (Comments on Novelty/Model System for Author):

Genome-wide CRISPR screening tech, identification of PKMYT1 as a novel potential therapeutic target of PDAC; Multiple PDAC cell models, PDAC cell line-derived xenograft and clinically relevant patient-derived xenograft models were used.

Referee #3 (Remarks for Author):

The authors have adequately addressed my comments.

***** Reviewer's comments *****

Referee #2 (Remarks for Author):

The authors have addressed most of my concerns. Please find some minor remaining points:

>> Which currently given cytotoxic does the RP-6306 drug synergise with most?

> gemcitabine

I meant to ask whether the authors could show to which cytotoxic such synergy could be found. Have they also tested 5-FU, nab-paclitaxel, oxaliplatin, irinotecan?

We thank the reviewer for the helpful comment. We have tested gemcitabine (MCE #HY-17026) and paclitaxel (Sangon Biotechnology #A601183) and found that PKMYT1 depletion sensitizes pancreatic cancer cells to gemcitabine but not to paclitaxel (Response Figure 1). One possible explanation is that paclitaxel is too sensitive in the tested cell lines to show extra additive effects. Because the detailed mechanisms merit further investigation and the paclitaxel data is preliminary. Therefore, we have not included the paclitaxel data in the manuscript. 5-Fluorouracil, oxaliplatin and irinotecan have not been tested.

Response Figure 1. PKMYT1 depletion sensitized pancreatic cancer cells to gemcitabine (left) but not to paclitaxel (right).

Do note that something strange happened to Figure 1C.

We have updated the related parts. Thank you!

Figure 2B could perhaps just be mentioned in the text?

Former Figure 2B has been removed and mentioned in the text as follow: Immunohistochemical (IHC) staining with validated antibodies against PKMYT1 demonstrated high PKMYT1 expression (36%) on a human tissue microarray consisting of 75 patients with PDAC

Can the authors find a way to simplify the numerous significance values in Figure 4E? Perhaps a graphical way to plot these numbers?

We have plotted these numbers in a simple way in the Figure 4E.

It would help to harmonise font sizes across panels and figures.

We have adjusted the font sizes. Thank you!

14th Mar 2024

Dear Dr. Wang,

Thank you for sending your revised files. I am pleased to inform you that your manuscript is accepted for publication and is now being sent to our publisher to be included in the next available issue of EMBO Molecular Medicine.

In the checklist, please note that I have removed the text from the section "Experimental animals/animal observed in or captured from the field", as I don't think it applies to your study. Please let me know immediately if you don't agree.

Yours sincerely,

Lise Roth
